# Disease-associated mutations within the yeast DNAJB6 homolog Sis1 slow conformer-specific substrate processing and can be corrected by the modulation of nucleotide exchange factors

Ankan K. Bhadra [1], Michael J. Rau [2], Jil A. Daw[3], James A. J. Fitzpatrick [1,2,4], Conrad C. Weihl[3] & Heather L. True [1] ✉

Molecular chaperones, or heat shock proteins (HSPs), protect against the toxic misfolding and aggregation of proteins. As such, mutations or deficiencies within the chaperone network can lead to disease. Dominant mutations within DNAJB6 (Hsp40)—an Hsp70 co-chaperone—lead to a protein aggregation-linked myopathy termed Limb-Girdle Muscular Dystrophy Type D1 (LGMDD1). Here, we used the yeast prion model client in conjunction with in vitro chaperone activity assays to gain mechanistic insights into the molecular basis of LGMDD1. Here, we show how mutations analogous to those found in LGMDD1 affect Sis1 (a functional homolog of human DNAJB6) function by altering the structure of client protein aggregates, interfering with the Hsp70 ATPase cycle, dimerization and substrate processing; poisoning the function of wild-type protein. These results uncover the mechanisms through which LGMDD1-associated mutations alter chaperone activity, and provide insights relevant to potential therapeutic interventions.

Limb-girdle muscular dystrophies (LGMDs) are a genetically heterogeneous family of muscle disorders that are either autosomal dominant or recessive[1]. Although most recessive LGMDs are characterized by a loss-of-function, the mechanistic nature of dominantly inherited LGMDs is unclear. These late onset degenerative myopathies are unified by similar myopathologies that include myofibrillar disorganization, impaired protein degradation, and the accumulation of protein inclusions that contain structural muscle proteins such as desmin and α-actinin and RNA binding proteins such as TDP-43[2–4]. This toxic misfolding and aggregation of proteins is protected by the activity of molecular chaperones, or heat shock proteins (HSPs). As such,

mutations in these chaperones can lead to diseases termed "chaperonopathies". One such example is Limb-Girdle Muscular Dystrophy Type D1 (LGMDD1), caused by mutations in DNAJB6 (Hsp40), an Hsp70 co-chaperone[4]. The originally identified LGMDD1 disease mutations are present within the 12 amino acid stretch of the glycine/phenylanine (G/F) rich domain[3–7]. DNAJB6 is expressed ubiquitously and participates in protein folding and disaggregation[8–11]; however, its role in skeletal muscle protein homeostasis is unknown. In addition, DNAJB6 client proteins and DNAJB6 chaperone interactions in skeletal muscle are not known. Fortunately, DNAJB clients are well-characterized in yeast and thereby afford a model system to study the effect of disease-

[1]Department of Cell Biology and Physiology, Washington University School of Medicine, 660 South Euclid Avenue, Campus Box 8228, St. Louis, MO 63110, USA. [2]Washington University Center for Cellular Imaging (WUCCI), Washington University School of Medicine, St. Louis, MO, USA. [3]Department of Neurology, Hope Center for Neurological Diseases, Washington University School of Medicine, St. Louis, MO, USA. [4]Department of Neuroscience, Washington University School of Medicine, St. Louis, MO, USA. ✉e-mail: heather.true@wustl.edu

causing mutants. Moreover, the understanding of DNAJB function within the yeast chaperone network is more complete than in skeletal muscle. Previously, we utilized a transdisciplinary approach to ascertain the functionality of LGMDD1-associated mutants in model systems[12]. Here, we generated homologous DNAJB6 LGMDD1 G/F domain mutations in the essential yeast DNAJ protein Sis1 (DNAJB6-F93L (Sis1-F106L), DNAJB6-N95L (Sis1-N108L), DNAJB6-D98Δ (Sis1-D110Δ), and DNAJB6-F100I (Sis1-F115I)). Our goal is to accelerate our understanding of mutant DNAJB6 dysfunction in LGMDD1 and thus facilitate therapeutic target identification as well as to gain insight into the relationship between protein quality control and myopathy.

The Hsp70/DNAJ machinery is vital to the protein quality control network. The Hsp70 machine works in an ATP-dependent manner to act on client proteins through a cycle of regulated binding and release[13]. Client specificity of the Hsp70 machine is modulated by DNAJ proteins (Hsp40s). By dictating client specificity, DNAJ proteins have been described as the primary facilitators of the cellular protein quality control system and play a pivotal role in determining the fate of a misfolded protein—whether is refolded or degraded[13]. A variety of disease-associated misfolded proteins have been shown to interact or colocalize with DNAJ family members[8,14]. These effects are generally, although not exclusively, dependent upon cooperation with Hsp70[8,13]. Strikingly, previous work from our lab suggests that the LGMDD1 mutants not only show substrate specificity, but also show conformation-specific effects[12,15]. These results were obtained through analysis of LGMDD1 mutants in the DNAJ protein Sis1, which has well-known yeast prion protein clients. Unlike mammalian prions, yeast prions are non-toxic, but phenotypic and biochemical assays enable rapid detection of [PRION+] cells[16]. Yeast chaperones Hsp104, Hsp70 (Ssa1), and the Hsp40 (Sis1) regulate prion propagation by acting on prion protein aggregates[17]. Alterations in chaperone level or function result in a failure to promote prion propagation[18,19].

One of the most interesting features of prions is the existence of prion strains. Prion strains are distinct self-propagating protein aggregate structures that cause changes in transmissibility and disease pathology with the same aggregating protein[20]. Yeast prion strains differ from each other based on phenotype, the ratio of soluble to aggregated protein, and their ability to propagate the prion[20]. Previously, we found that homologous LGMDD1 mutations in Sis1 appear to reduce functionality, as determined by changes in their ability to modulate the aggregated state of select yeast prion strains[12]. We then assessed the effect of these mutants in mammalian systems, including mouse models, and LGMDD1 patient fibroblasts, where we analyzed the aggregation of TDP-43, an RNA binding protein with a prion-like domain that is a marker of degenerative disease including LGMDD1. DNAJB6 mutant expression enhanced the aggregation and impaired the dissolution of nuclear stress granules containing TDP-43 following heat shock[12]. Recently, three novel pathogenic mutations associated with LGMDD1 have been identified within the J domain of DNAJB6[21]. Interestingly, we found that homologous mutations in the Sis1 J domain differentially alter the processing of specific yeast prion strains, as well as a non-prion substrate[15].

Here, we used the yeast prion model system, and in vitro chaperone activity assays to determine how LGMDD1 homologous missense mutations in the Sis1 G/F domain alter chaperone function with and without Hsp70. We found that LGMDD1 mutants inhibit the Hsp70 ATPase cycle function in a client-conformer-specific manner. Moreover, both prion propagation and luciferase refolding activity were enhanced in mutant strains by either deleting the NEF (Sse1) or by using an Sse1-mutant, indicating that fine-tuning of substrate processing can rescue the mutant defects. To our knowledge, this is the first mechanistic insight describing the effect of LGMDD1 mutants on Hsp70/40 ATPase cycle. Additionally, these results suggest that the development of a titrated approach using specific inhibitors of the

Hsp70/DNAJ cycle is a potential therapeutic strategy for this class of myopathy-associated chaperonopathies.

## Results

### LGMDD1-associated homologous G/F domain mutants in Sis1 have variable substrate processing efficiency

In previous studies, we analyzed the functionality of LGMDD1 mutants in Sis1 using two yeast prion proteins that require Sis1 for propagation: Rnq1 (which forms the [RNQ+] prion) and Sup35 (which forms the [PSI+] prion). We found that LGMDD1 mutants had conformer-specific processing defects, in that they promoted the propagation of some prion strains but not all[12]. This suggests that the mutant chaperones may recognize their normal clients in some conformations but not others. Alternatively, the mutant chaperones may recognize clients but not be able to effectively process the misfolded protein. Hsp40 client processing for refolding requires multiple steps: binding, dimerization, binding to Hsp70, and nucleotide exchange-stimulated client release. We evaluated the function of LGMDD1 mutants as compared to wild type (WT), by purifying these proteins (Supplementary Fig. 1) and using them in assays that address these steps.

One way to determine substrate processing of a chaperone that interacts with amyloidogenic proteins is to evaluate the kinetics of aggregation of client proteins in vitro. We assessed amyloid formation of Rnq1 in the presence of Sis1-WT and LGMDD1 mutants (G/F domain mutants Sis1-F106L, Sis1-N108L, Sis1-D110Δ and, Sis1-F115I). Rnq1 is a well-known substrate of Sis1[22] and has been reported to form higher-order aggregates in vitro[23]. Aggregation of Rnq1 was monitored by the enhanced fluorescence emission of the dye Thioflavin T (ThT), which is used as a marker for cross β sheet conformation of amyloid fibrils[24]. During the initial phase of incubation (lag phase), natively disordered protein monomer did not show any change in fluorescence intensity of ThT (Fig. 1a). This was followed by an increase in ThT fluorescence intensity, indicative of the formation of fibrils. Finally, the fluorescence intensity of the dye achieved a plateau (stationary phase). The lag phase was increased in the presence of Sis1-WT as compared to Rnq1 without chaperone (Fig. 1a), indicating that the time required for productive nucleation of unseeded Rnq1 was longer in the presence of chaperone. However, the lag phase of Rnq1 fiber formation was shorter with LGMDD1 mutants than Sis1-WT but longer than that of Rnq1 alone (Fig. 1a). The overall fibril formation rate was also higher in the presence of the mutants as compared to Sis1-WT (Fig. 1a). The time to 50% ThT is shown in the inset (Fig. 1a). We also performed seeded kinetic assays with Rnq1 fibers formed at 18, 25, and 37 °C. We found that fiber elongation was faster (in the absence of chaperone) with seeded Rnq1 at 18 °C, than at 25 °C or 37 °C (Supplementary Fig. 2a–c). Similarly, in the presence of Sis1-WT, the difference in fiber elongation using Rnq1 seeds formed at the three temperatures was negligible (Supplementary Fig. 2d–f). This suggests that the primary impact of Sis1 on Rnq1 fiber formation is in the formation of amyloid-competent conformers (nucleation).

We then hypothesized that, if the primary chaperone effect is to alter the nucleation of the amyloidogenic protein, the fibers formed with and without chaperone could be structurally distinct. As such, we assessed the morphology of Rnq1 fibrils formed at 18 °C with and without chaperone by transmission electron microscopy (TEM). TEM images show that Rnq1 forms long, elongated, branched fibrils in the absence of chaperone (Fig. 1b). By contrast, Rnq1 fibers formed in the presence of Sis1-WT were short and appeared immature or perhaps bound to chaperone (Fig. 1b). This suggests that Sis1-WT might only delay fibril formation rather than preventing it. In the presence of LGMDD1 mutant (Sis1-F115I), very few fibers were visible (Fig. 1b), and they were mostly distorted and appeared to be small oligomeric species. Taken together with our ThT findings, these data suggest that different ThT binding sites are present within these morphologically

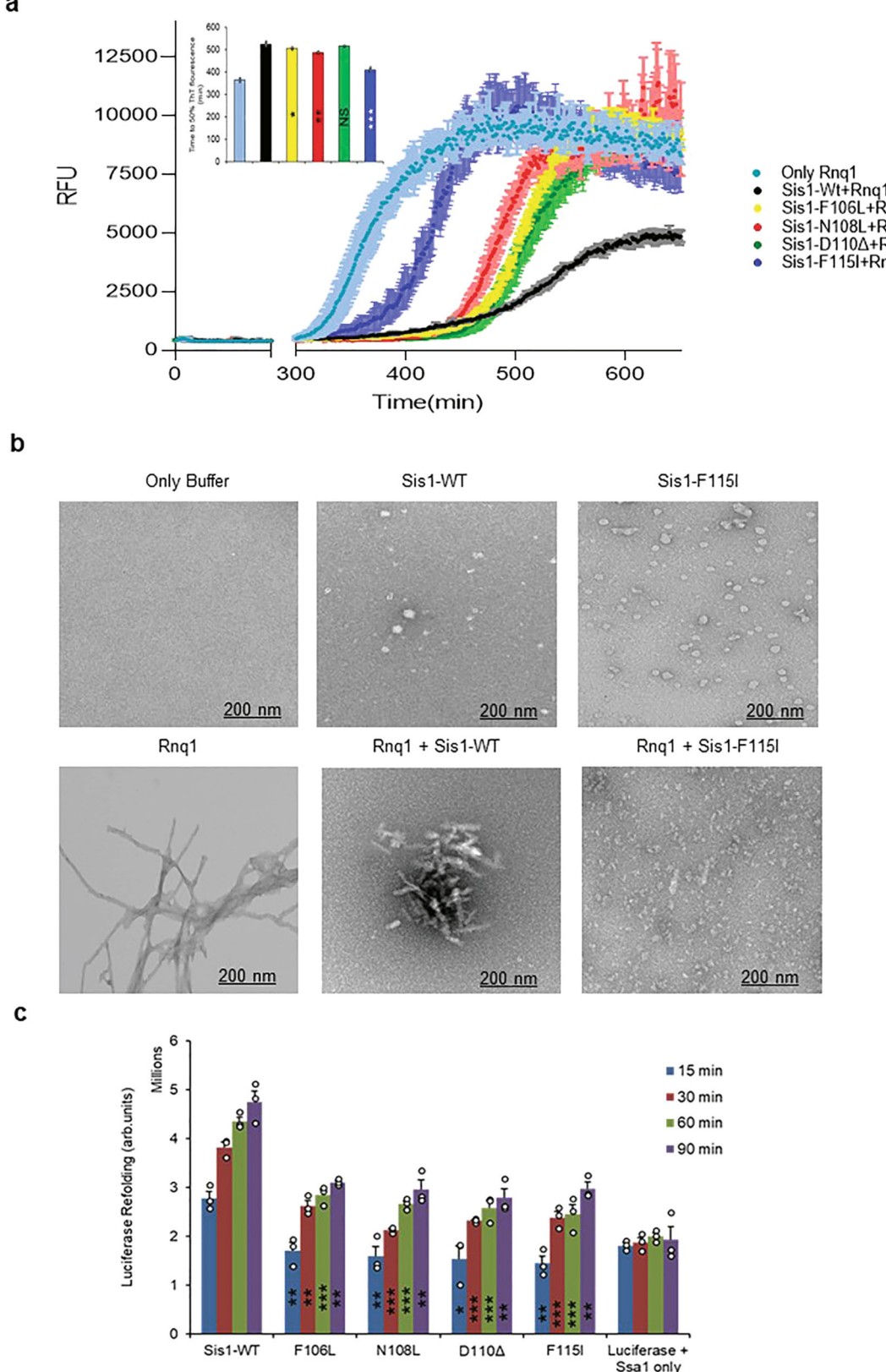

distinct Rnq1 aggregates. Additionally, these data suggest that the interaction of LGMDD1 mutants with client alter client conformation in a manner that is distinct from that of wildtype chaperone.

While the interaction with prion substrate showed that LGMDD1 mutants alter amyloid formation, DNAJ proteins typically act in conjunction with Hsp70 to process substrates. We then asked

whether the LGMDD1 mutants could promote substrate refolding in the presence of co-chaperones. Previously Aron et al. showed that Sis1 lacking the G/F domain (Sis1ΔG/F) is defective in chaperone activity and partially inhibits the ability of Sis1-WT to facilitate folding of denatured luciferase protein[25]. Therefore, to test the ability of LGMDD1 mutants to function in substrate refolding, we heat-

**Fig. 1 | LGMDD1 G/F domain mutants show variability in substrate processing. a** Kinetics of Rnq1 fibrillation in the presence of unseeded Rnq1 only (cyan blue), Sis1-WT (black), Sis1-F106L (yellow), Sis1-N108L (red), Sis1-D110Δ (green), or Sis1-F115I (blue) measured by ThT fluorescence assay. *Inset*: - Time to 50% ThT fluorescence was calculated by fitting the graph using the EC50 (Y = Bottom + (Top-Bottom)/(1 + 10^((lnEC50-X)*HillSlope)) equation in Graph pad prism. RFU is Relative fluorescence unit. Values shown are mean ± SEM, *n* = 3 biologically independent samples. For (**a**; inset) each LGMDD1 mutant was compared with Sis1-WT across and ***p < 0.00019 for Sis1-F115I, **p < 0.00446 for Sis1-N108L, *p < 0.0489 for Sis1-F106L, NS = 0.25297 (non-significant) for Sis1-D110Δ values are reported for unpaired, two-sided *t*-test. **b** The morphology of amyloid fibers formed from Rnq1

at 18 °C in vitro in the presence and absence of Sis1-WT and/or LGMDD1 mutant were imaged by TEM. Experiments were performed in triplicate, representative images shown. The scale bar represents 200 nm. **c** Refolding activity of heat-denatured luciferase in the presence of LGMDD1 mutants. Luciferase along with Ssa1-WT and Sis1-WT/mutants was incubated at 42 °C for 10 min to heat denature luciferase. At various time points, activity was measured by a luminometer after adding substrate. Values shown are mean + SEM, *n* = 3 biologically independent samples. For **c**; each LGMDD1 mutant was compared with Sis1-WT across all time-points and ***p < 0.001, **p < 0.01, and *p < 0.05 values are reported for unpaired, two-sided *t*-test. Source data are provided as a Source data file.

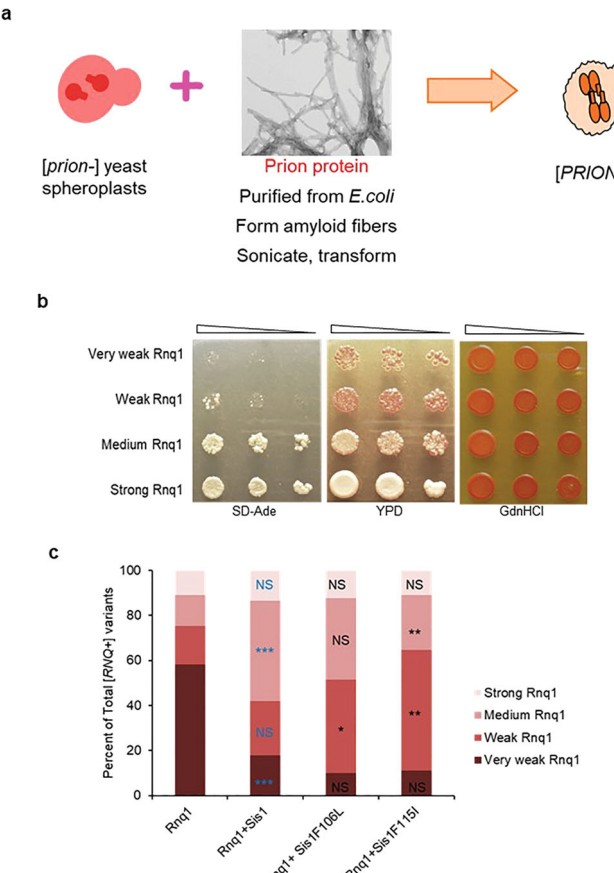

**Fig. 2 | Phenotypic distribution of [*RNQ*+] strain alters with LGMDD1 G/F domain mutants. a** Cartoon showing transformation of Rnq1 amyloid fibers formed at 18 °C into [*prion*-/*rnq*–] 74-D694 yeast cells induces strains of the [*PRION*+/*RNQ*+] prion. RRP was used to assess [*RNQ*+] phenotype. [*RNQ*+] prion strains co-aggregate with RRP and cause different phenotypes; colony color on YPD and growth on SD-ade medium are indicative of different levels of suppression of the *ade1-14* premature stop codon to produce Ade1 and represent different [*RNQ*+] strains. Curability by growth and color on medium containing GdnHCl was used to determine prion-dependence of phenotypes. The transformants from five separate experiments for each sample set were picked and >200 colonies for each set were scored as very weak, weak, medium, or strong [*RNQ*+] **b** and graphed for statistical analysis **c**. In panel **c**, the blue color indicates the comparison between Rnq1 (alone) and Rnq1 with Sis1-WT; NS = 0.197 for strong [*RNQ*+], ***p < 0.00023 for medium [*RNQ*+], NS = 0.357 for weak [*RNQ*+], ***p < 0.00025 for very weak [*RNQ*+]. Black color indicates the comparison between Sis1-WT and LGMDD1 mutants (F106L and F115I); (NS = 0.298 and 0.269) for strong [*RNQ*+], (NS = 0.165 and **p < 0.008) for medium [*RNQ*+], (*p < 0.022 and **p < 0.008) for weak [*RNQ*+], (NS = 0.231 and 0.409) for very weak [*RNQ*+]. *p*-values are reported for unpaired, two-sided *t*-test. Source data are provided as a Source data file.

denatured firefly luciferase and then monitored refolding in the presence or absence of chaperones. We found that the LGMDD1 mutants were all compromised in their ability to refold luciferase (Fig. 1c, see also controls Supplementary Fig. 2g). Taken together, these results indicate that the LGMDD1 mutants are defective in substrate processing.

**LGMDD1-associated G/F domain mutants change the formation of in vitro formed [*RNQ*+] prion strains**

Given the striking differences in fibril formation kinetics and TEM images, we wanted to determine whether LGMDD1 mutants alter the formation of Rnq1 amyloid structures using a more sensitive and quantitative method that is uniquely available in the yeast prion system. This entails infecting yeast with amyloid generated in vitro and assessing the resulting [*RNQ*+] prion strains. Previously, we found that amyloid fibers of Rnq1PFD (prion-forming domain) formed at different temperatures resulted in the generation of different prion strains[26].

We hypothesized that Rnq1 amyloid structure, and resulting [*RNQ*+] strain distribution, would differ in the presence of Sis1 and LGMDD1 mutants. We formed Rnq1 fibers at 18 °C in the presence and absence of Sis1-WT and two LGMDD1 mutants. We transformed fibers formed in the presence or absence of chaperone into cells expressing [*RNQ*+] reporter protein (RRP) that did not have the prion ([*rnq*-]) (Fig. 2a and Supplementary Fig. 3a). We developed and utilized this reporter strain as a phenotypic indicator of [*RNQ*+] prion strain propagation[26]. Yeast strains expressing the RRP reporter display different levels of *ade1-14* nonsense suppression in [*RNQ*+] variants (Supplementary Fig. 3b). By assessing colony color (white, light pink, or dark pink) as well as growth on selective medium (SD-Ade), we scored [*RNQ*+] strains (Fig. 2b and Supplementary Fig. 3b) that resulted after fiber infection. Lighter colony color and more growth on SD-Ade medium was scored as a stronger [*RNQ*+] strain, whereas darker colony color and less growth on selective medium was scored as a weaker [*RNQ*+] strain (see "Methods" for more information). [*RNQ*+] strains were also verified for the prion-specific trait of curability on medium containing guanidine hydrochloride (Fig. 2b). To quantify the difference in [*RNQ*+] strains formed in the presence of LGMDD1 mutants, we counted the phenotype of infected colonies from five different transformation sets for each sample (Fig. 2b). With Rnq1 alone, ~58% of cells showed very weak [*RNQ*+] phenotypes. This fraction of very weak [*RNQ*+] was reduced to less than 20% in the presence of Sis1-WT and was further reduced in the presence of the LGMDD1 mutants. There was a significant increase in the medium [*RNQ*+] strain phenotype (44% +/− 4.5%) in the presence of Sis1-WT as compared to Rnq1 alone (14% +/− 1.4%). When we compared [*RNQ*+] strain distribution between Sis1-WT and LGMDD1 mutants, we found that the proportion of weak Rnq1 strains was significantly increased with both F106L and F115I (~20% in Sis1-WT vs. 50% in F106L and 65% in F115I). Similarly, the population of medium [*RNQ*+] strain was significantly decreased in F115I as compared to Sis1-WT (~20% in F115I vs. 45% in Sis1-WT). The distribution of other [*RNQ*+] strains, strong and very weak, with LGMDD1 mutants were similar to Sis1-WT.

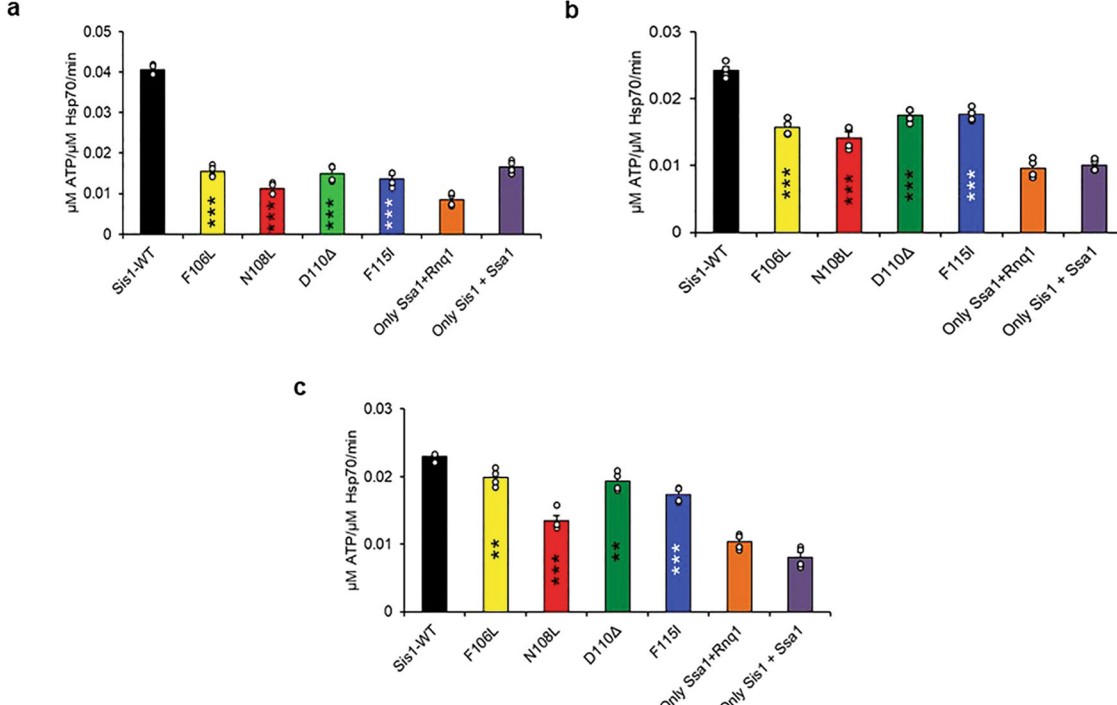

**Fig. 3 | Stimulation of ATPase activity of Ssa1 by LGMDD1 G/F domain mutants is client-conformation specific.** Stimulation of Ssa1 ATPase activity in the presence of Rnq1 seeds formed at **a** 18 °C, **b** 25 °C, and **c** 37 °C. Ssa1 (1 μM) in complex with ATP (1 mM) in the presence of Sis1-WT (black) or Sis1-mutants (Sis1-F106L (yellow), Sis1-N108L (red), Sis1-D110Δ (green), or Sis1-F115I (blue)) (0.05 μM). The fraction of ATP converted to ADP was determined. For **a**–**c**, a total of 10% seeds were used in a reaction. In all cases, only Ssa1 with Rnq1 (orange), Sis1 with Ssa1 (dark purple) and were used as controls. For **a**–**c**, Sis1-WT was compared with LGMDD1- mutants. Data are presented as mean values + SEM, *n* = 4 biologically independent samples. For **a** (left to right); ***p < 1.29e−07, ***p < 6.29e−08, ***p < 4.78e−07, ***p < 2.41e−07; for **b** (left to right); ***p < 4.64e−05, ***p < 6.31e−05, ***p < 0.0001, ***p < 0.0001; and for **c** (left to right); **p < 0.004, ***p < 2.63e−05, **p < 0.002, ***p < 9.59e−05. *p*-values are reported for unpaired, two-sided *t*-test. Source data are provided as a Source data file.

Thus, the difference in distribution of [*RNQ*+] strains generated in the presence of LGMDD1 mutants suggest that amyloid structures can be altered by mutant chaperones.

### LGMDD1-associated homologous mutants in Sis1 (Hsp40) alters its ability to function with Ssa1 (Hsp70) efficiently

Hsp40s target substrates to their Hsp70 partners and regulate the ATPase activity and substrate binding of the Hsp70[13]. The recognition of substrates depends on their conformation, and it has been suggested that much of the Hsp40 conformation-dependent recognition is dependent on the G/F domain[22,27–29]. Additionally, in our previous work, we found client-conformer specific defects in vivo with the LGMDD1 mutants[12]. Therefore, we asked whether the conformation of the client impacted the function of the mutants. We measured the ability of Sis1-WT and LGMDD1 mutants to stimulate the ATPase activity of Ssa1 in the presence and absence of different Rnq1 protein conformers. We standardized the Rnq1 monomer concentration and performed a phosphate standard curve with each assay (Supplementary Fig. 4a). Notably, there was no difference between Sis1-WT and LGMDD1 mutants stimulated Ssa1 ATP hydrolysis rate in the absence of any client protein (Supplementary Fig. 4b) or in the presence of Rnq1 monomer (Supplementary Fig. 4c). This may be due to the fact that "denatured monomer" presents a variety of epitopes that can be recognized by chaperone whereas the fibers likely have fewer sites for recognition. However, the stimulation of Ssa1 ATP hydrolysis rate was significantly reduced with the mutants in the presence of Rnq1 seeds formed at 18, 25, and 37 °C (Fig. 3a–c). Taken together, these data suggest that the different Rnq1 conformers afford differences in Sis1-mediated Hsp70 ATPase stimulation and the LGMDD1 mutants all show functional deficiencies.

### LGMDD1 G/F domain mutants alters both substrate and Hsp70 binding

The initial step in this chaperone cycle relies on the ability of Hsp40 to bind to client. Thus, we wanted to test the physical and functional interactions of LGMDD1 mutants with our Rnq1 and luciferase substrates. We performed a binding assay utilizing a method previously used to study interactions between the *E. coli* DnaJ and substrates[30]. We found that LGMDD1 mutants show significantly reduced binding to denatured Rnq1 (similar substrate used for Supplementary Fig. 4c) and luciferase substrates as compared to Sis1-WT (Fig. 4a, b) at low concentrations of substrate.

A key function for Hsp40s is stimulating Hsp70s. Our previous work, suggests that the deleterious effect of the DNAJB6 G/F domain mutants is Hsp70-dependent[31]. Thus, the LGMDD1 mutants might be altered in their productive association with Hsp70. These mutants may affect the cycle by either reducing Hsp70 binding, sequestering Hsp70, or they may be hyperactive and alter the refolding process. Therefore, we performed binding assays to determine whether there was a productive association between the LGMDD1 mutants and Ssa1. We found that the interaction between LGMDD1 mutants and Ssa1 was significantly reduced both in the absence (Fig. 4c) and presence of client proteins Rnq1 (Fig. 4d) and luciferase (Supplementary Fig. 5). This indicates that the reduced interaction between the mutants and Ssa1 is client-independent. Thus, the decrease in Hsp70 ATPase activity with LGMDD1 mutants (Fig. 3a–c) could, at least in part, be due to reduced Hsp70 binding.

### LGMDD1 G/F domain mutants show reduced dimerization efficiency

Most of the canonical class I and class II J-proteins exist as oligomers[13]. The efficient function of Sis1 requires the ability of the

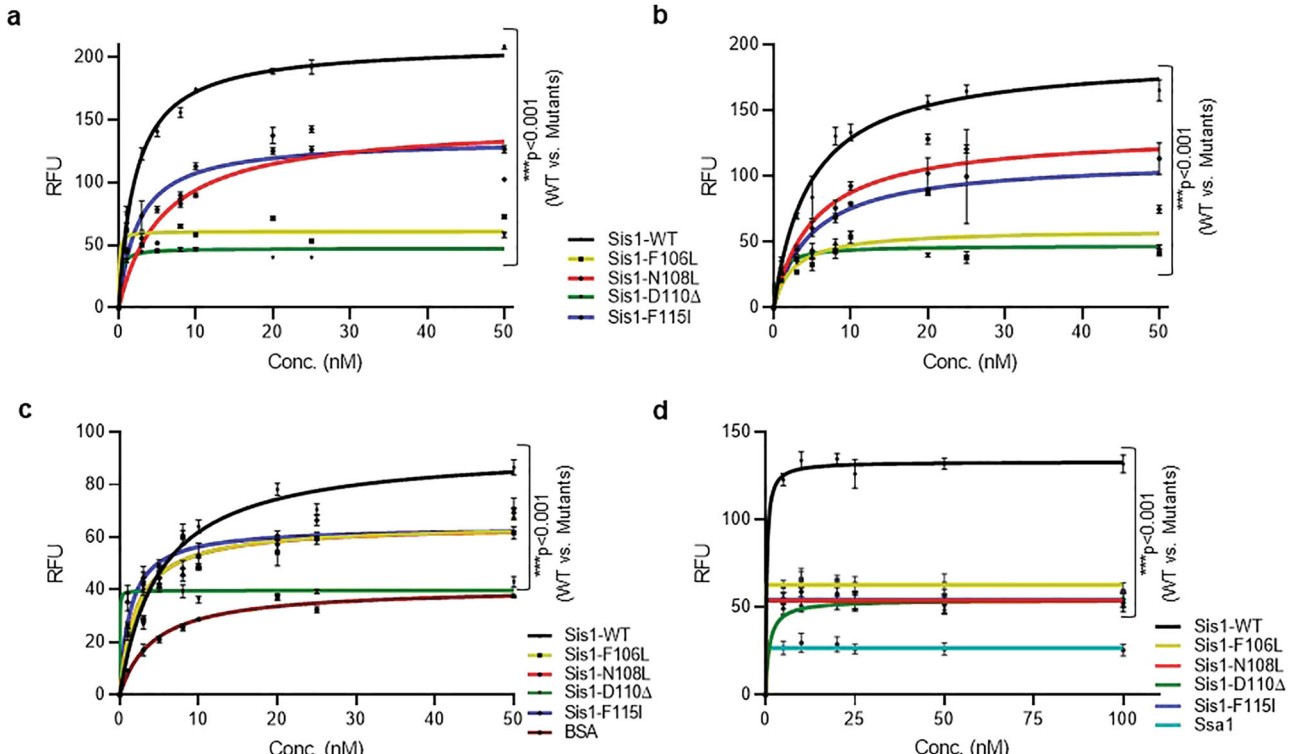

**Fig. 4 | Sis1 binding to both substrate and Hsp70 is compromised in the presence of LGMDD1 G/F domain mutants.** Binding of purified Sis1-WT (black), Sis1-F106L (yellow), Sis1-N108L (red), Sis1-D110Δ (green), or Sis1-F115I (blue) to denatured Rnq1 **a** and luciferase **b**. Rnq1 (400 ng) and luciferase (100 ng) were immobilized in microtiter plate wells and dilutions of purified Sis1-WT and Sis1-mutants (0, 1, 3, 5, 8, 10, 20, 25, and 50 nM) were incubated with each substrate. The amount of Sis1 retained in the wells after extensive washings was detected using a Sis1 specific antibody. For **c**; Ssa1 (200 nM) was immobilized in microtiter plate wells and dilutions (0, 1, 3, 5, 8, 10, 20, 25, and 50 nM) of purified Sis1-WT (black), and Sis1-mutants (Sis1-F106L (yellow), Sis1-N108L (red), Sis1-D110Δ (green), or Sis1-

F115I (blue)) or BSA (brown) as a control, were incubated with it. Bound Sis1 was detected using an αSis1 antibody. **d** Denatured Rnq1 were premixed with Sis1-WT/ mutants and immobilized in microtiter plate wells and dilutions of Ssa1-WT (0–100 nM) were incubated with it. Bound Ssa1-WT was detected using an αSsa1 antibody. Only Ssa1 (cyan blue) was used as a control for **d**. Data represented as mean values ± SEM, $n = 3$ biologically independent samples. For **a**–**d**, Sis1-WT was compared to LGMDD1 mutants and $***p < 0.001$ values are reported for unpaired, two-sided $t$-test. RFU stands for Relative fluorescence unit. Source data are provided as a Source data file.

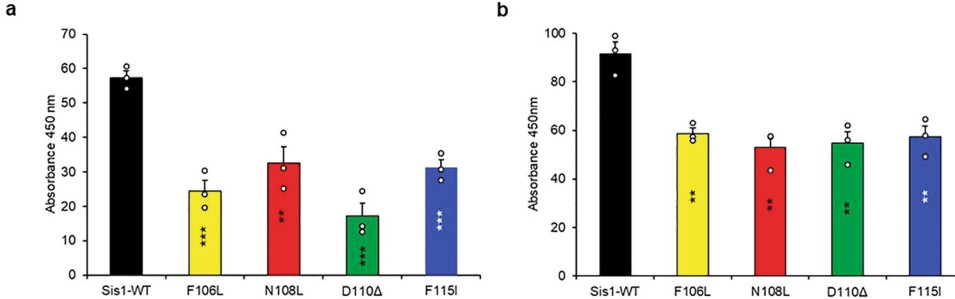

**Fig. 5 | LGMDD1 G/F domain mutants showed reduced dimerization. a** For homodimerization, uncleaved His-tagged Sis1-WT and mutants (20 nM) were added into non-His-tagged (cleaved) Sis1-WT and mutants (200 nM) and adsorbed in microtiter plate wells. **b** For heterodimerization, uncleaved His-tagged Sis1-WT (20 nM) was added to cleaved mutants (200 nM) and adsorbed in microtiter plate wells. In both **a**, **b**, following adsorption, ELISA was performed using an anti-His

antibody. Values shown are mean + SEM, $n = 3$ biologically independent samples. For **a**, **b**; all LGMDD1 mutants were compared with Sis1-WT; for F106L ($***p < 0.0008$ and $**p < 0.003$), for N108L ($**p < 0.008$ and $**p < 0.004$), for D110Δ ($***p < 0.0006$ and $**p < 0.005$), and for F115I ($**p < 0.0009$ and $**p < 0.006$) values are reported for unpaired, two-sided $t$-test. Source data are provided as a Source data file.

protein to form dimers. In fact, Sis1 (1–337aa), which lacks the dimerization motif (Sis1-ΔDD), exhibited severe defects in chaperone activity, but could regulate Hsp70 ATPase activity[32]. Sha et al. proposed that the Sis1 cleft formed in dimers functions as a docking site for the Hsp70 peptide-binding domain and that this interaction facilitates the transfer of peptides from Sis1 to Hsp70[33]. Thus, in order to further evaluate the chaperone activity of the these mutants, we set out to measure their dimerization efficiency. We

performed binding assays to determine the relative competence of both the self-association of the mutants as well as their association with Sis1-WT. We found that there was a significant decrease in self-association of all the mutants as compared to Sis1-WT (Fig. 5a). Sis1-WT lacking the dimerization domain (Sis1-ΔDD) was used a control and showed reduced dimerization as compared to Sis1-WT (Supplementary Fig. 6). We also found that their ability to bind to Sis1-WT was significantly reduced (Fig. 5b), indicating a possible

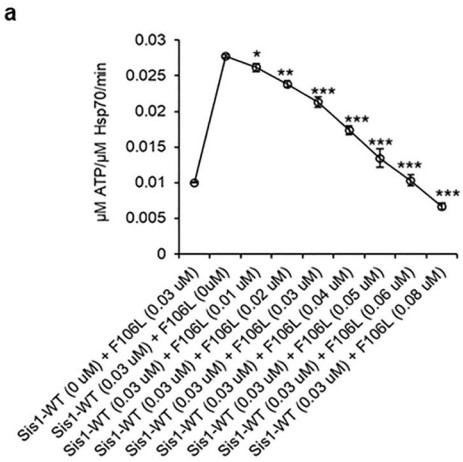

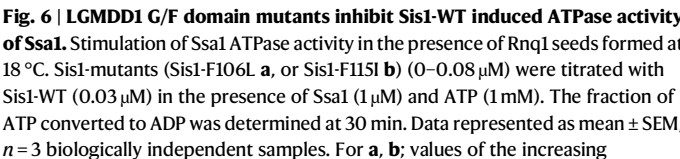

**Fig. 6 | LGMDD1 G/F domain mutants inhibit Sis1-WT induced ATPase activity of Ssa1.** Stimulation of Ssa1 ATPase activity in the presence of Rnq1 seeds formed at 18 °C. Sis1-mutants (Sis1-F106L **a**, or Sis1-F115I **b**) (0–0.08 μM) were titrated with Sis1-WT (0.03 μM) in the presence of Ssa1 (1 μM) and ATP (1 mM). The fraction of ATP converted to ADP was determined at 30 min. Data represented as mean ± SEM, $n = 3$ biologically independent samples. For **a**, **b**; values of the increasing

concentration of LGMDD1 mutants (0.01–0.08 μM) were compared with Sis1-WT (0.03 μM) alone. For **a** (top to bottom); $*p < 0.049$, $**p < 0.001$, $***p < 0.0008$, $***p < 0.0001$, $***p < 0.0003$, $***p < 2.18e{-}05$, $***p < 2.62e{-}06$. For **b** (top to bottom); $**p < 0.007$, $*p < 0.012$, $*p < 0.01$, $***p < 0.0008$, $***p < 0.0001$, $***p < 6.4e{-}05$, $***p < 2.11e{-}05$. $p$-values are reported for unpaired, two-sided $t$-test. Source data are provided as a Source data file.

explanation for the reduction in chaperone activity observed with these mutants.

## LGMDD1 G/F domain mutants inhibit Sis1-WT induced Ssa1 ATPase activity by reducing its dimerization and substrate binding efficiency

Since LGMDD1 has been described as an autosomal dominant disease, we set out to test the effect of LGMDD1 mutants on the Sis1-WT in functional assays. We performed ATPase assays with mixtures of Sis1-WT and LGMDD1 mutants F106L and F115I in the presence of Rnq1 fibers formed at 18 and 25 °C. We found a gradual decrease in the rate of ATP hydrolysis that correlated with titrating Sis1-WT with F106L (Fig. 6a and Supplementary Fig. 7a) and F115I (Fig. 6b and Supplementary Fig. 7b).

Next, we wanted to determine whether the mutants also inhibit the ability of wild-type protein to dimerize. We found that there was a concomitant decrease in dimerization of Sis1-WT with the addition of F106L (Supplementary Fig. 8a) and F115I (Supplementary Fig. 8b). We then asked whether the mutants also inhibit the ability of wild-type protein to bind substrates efficiently. The binding efficiency to Rnq1 (Supplementary Fig. 8c) and luciferase (Supplementary Fig. 8d) substrates were slightly but significantly reduced when Sis1-WT was used in equal proportion (1:1) with each of the LGMDD1 mutants F106L and F115I. These data provide mechanistic insight into the inhibition of Sis1-WT-induced Ssa1 ATPase activity in the presence of LGMDD1 mutants (Fig. 6).

## 7. Modulating Hsp40-Hsp70 cycle by either deleting or inhibiting nucleotide exchange factors (NEFs) can be beneficial with respect to LGMDD1 G/F mutant effect in vivo

The lifetime of the Hsp70/40:substrate complex is dependent upon nucleotide exchange. A key player in this is nucleotide exchange factors (NEFs) that stimulate ADP release. In yeast, cytosolic Hsp70 interacts with three NEFs homologous to human counterparts: Sse1/Sse2 (Hsp110), Fes1 (HspBP1), and Snl1 (Bag-1)[34,35]. Previously, we found that LGMDD1 mutants impair viability and prion propagation in yeast and these effects were rescued by reducing the association with Hsp70[31]. Thus, we decided to investigate whether deleting Hsp110 (Sse1) would have a similar rescuing effect. To assess the impact of NEFs, we asked whether their deletion would enhance the ability of LGMDD1 mutants to propagate [*RNQ*+] prion strains. We

used two established biochemical yeast prion assays that differentiate soluble and aggregated protein: well-trap and boiled gel assays. We found that the deletion of Sse1 partially rescues [*RNQ*+] prion propagation (Fig. 7a, b and Supplementary Fig. 9a). However, it was specific to Sse1, as the alteration of the other NEFs did not demonstrate such rescuing (Supplementary Fig. 9b). This was not surprising, as Sse1 is the principal NEF in yeast and performs 90% of the NEF activity in the cell[36].

In order to confirm that deletion of Sse1 does improve the [*RNQ*+] prion propagation, we assessed the effect of a well characterized Sse1 mutant, (Sse1-G233D). This mutation eliminates ATP binding by Sse1, and therefore binding to Ssa1, nearly completely. It therefore delays release of ATP from Ssa1 but only because it never binds Ssa1, and is essentially identical to strains lacking Sse1[37]. We performed boiled gel assays to examine the relative levels of soluble Rnq1 prion protein in [*RNQ*+] cells. Indeed, we found a decrease in soluble Rnq1 protein in LGMDD1 cells expressing Sse1-G233D (Supplementary Fig. 9c). This indicates restoration of [*RNQ*+] prion propagation in LGMDD1 mutants with NEF modulation.

To further understand the impact of Sse1 on the LGMDD1 disease-associated mutants, we assessed the refolding of a non-prion substrate (firefly luciferase; FFL). We utilized an in vivo refolding assay in which FFL is denatured in cells by heat shock and its subsequent refolding, which requires the Hsp40/Hsp70/Hsp104 chaperone machinery, is measured by activity[38,39]. We transformed the Sse1 and Δsse1 yeast cells carrying Sis1-WT, LGMDD1 mutants, and Sse1-G233D with GPD-lux vector for the expression of FFL. Since Hsp104 is required for efficient refolding of FFL, we used a Δhsp104 strain expressing FFL as a negative control. We found that the FFL refolding activity was significantly altered in Sse1 cells expressing F115I as compared to cells expressing Sis1-WT (Fig. 7c). This difference in FFL refolding activity between LGMDD1 mutants (F115I) and Sis1-WT was non-existent in Δsse1 yeast cells (Fig. 7c). However, we observed that the FFL refolding activity was marginally higher in Δsse1 yeast cells expressing F115I as compared to Sse1-WT cells expressing the same mutant protein (Fig. 7c). Interestingly, Δsse1 yeast cells co-expressing F115I and Sse1-G233D showed significant improvement in FFL refolding activity as compared to Sse1-WT yeast cells (Fig. 7c). Taken together, our results indicate that LGMDD1 mutants delay Hsp70 ATPase activity, possibly resulting in the increased load of aggregation-prone muscular proteins observed in LGMDD1 patients.

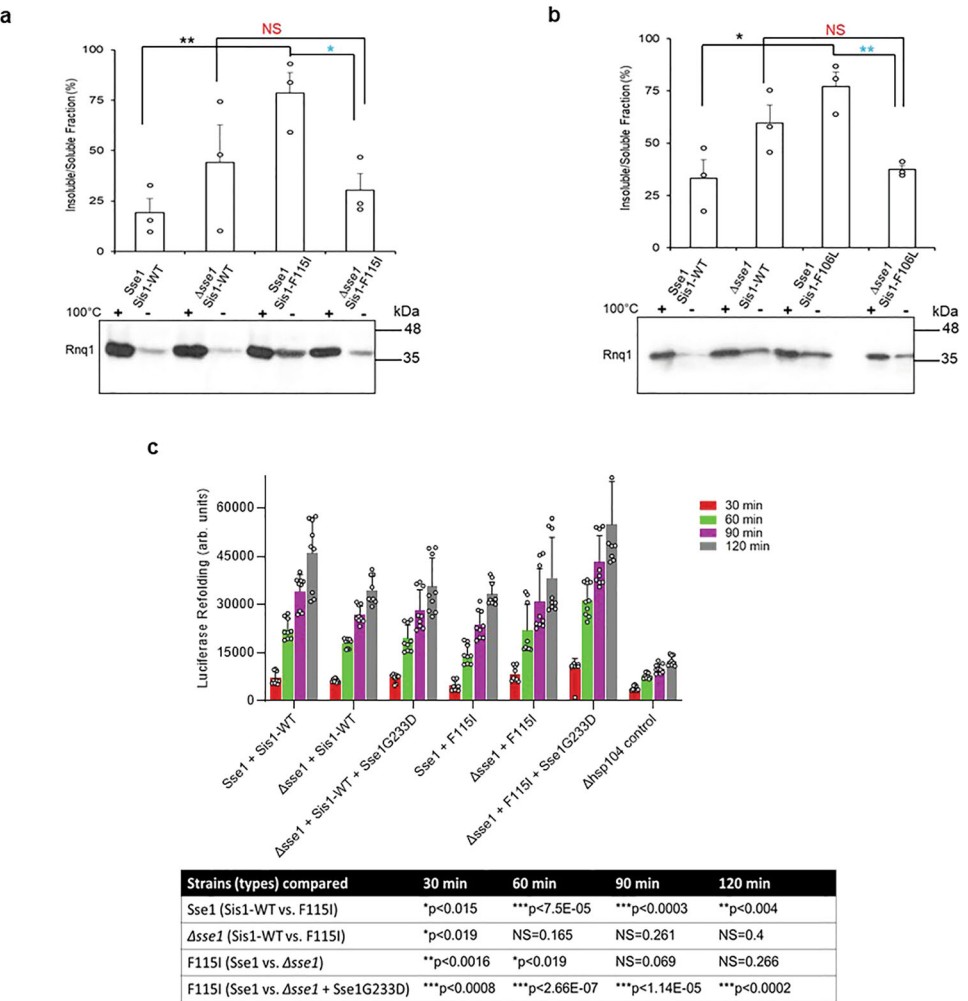

**Fig. 7 | Deletion or inhibition of Sse1 function rescues prion propagation of [*RNQ*+] and restored impaired FFL refolding. a, b** Deletion of Sse1 partially suppresses prion loss caused by LGMDD1 mutants. Analysis of prion propagation in Sse1 and *Δsse1* strains harboring s. d. medium [*RNQ*+], expressing either Sis1-WT or Sis1-F115I or Sis1-F106L by well-trap assay. Cell lysates were incubated at room temperature (−) or 100 °C and subjected to SDS-PAGE and western blot using an αRnq1 antibody. Rnq1 that is not sequestered in aggregates will enter the gel in the unboiled sample, which indicates destabilization of the [*RNQ*+] prion. The western blotting was done in triplicate and representative image is shown here. For **a, b**; the lines above the bars shows the samples which were compared and \*\*$p < 0.01$,

\*$p < 0.05$, NS (non-significant) values are reported for unpaired, two-sided *t*-test. **c** The refolding of firefly luciferase (FFL) was measured in the Sse1 and *Δsse1* yeast cells carrying Sis1-WT, LGMDD1 mutant (Sis1-F115I), and Sse1 mutant (Sse1-G233D) along with a plasmid expressing FFL. Yeast were normalized, treated with cycloheximide, and subjected to heat shock at 42 °C for 22 min, followed by recovery at 30 °C. Luminescence was measured at the indicated time points during recovery. Values shown are mean + SEM, *n* = 9 biologically independent samples. The table shows the samples that were compared at each time points. \*\*\*$p < 0.001$, \*\*$p < 0.01$, \*$p < 0.05$, NS (non-significant) values are reported for unpaired, two-sided *t*-test. Source data are provided as a Source data file.

## Discussion

Proper functioning of protein chaperones, including that of Hsp70/DNAJ, is important for the maintenance of muscle function[40] (Supplementary Fig. 10). Previously, we have shown that the LGMDD1 mutations in the G/F domain of DNAJB6 disrupt client processing in both a substrate- and conformation-specific manner[12]. Here, we delved into the mechanism underlying the defect in client processing shown by LGMDD1 G/F domain mutants. Our data show that the LGMDD1 mutants ability to stimulate Hsp70 ATPase activity was reduced as compared to the wild-type protein (Fig. 8a), which may result in general impairment of protein quality control and accumulation of protein inclusions in muscle. Analyzing the severity of the mutants, we find a significant difference in many steps in the Hsp40/70- cycle. No aspect of its functionality is unaffected by these disease causing G/F domain mutants (Fig. 8a). All mutants showed decreased binding to substrate (Fig. 4a, b), with D110Δ = F106L < F115 = N108L. Mutants also impact dimerization similarly with D110Δ = F106L < N108L = F115I (Fig. 5). The LGMDD1 mutants showed similar binding defects to Hsp70 (D110Δ

worse without client) (Fig. 4c). The mutants showed similar reduction in Hsp70 ATPase stimulation, but it was client conformation-dependent, as expected based on previous work[12] (Fig. 3a–c and Supplementary Fig. 4c). Finally, the mutants showed similar reduction in luciferase refolding efficiency (Fig. 1c).

We also took advantage of the yeast prion model system and its unique conformer:phenotype read-out. Prion proteins can form several unique prion variants (or strains) that have slight differences in their β-sheet structure that constitute distinct amyloid conformations[41]. Such different structures are presumably the underlying cause of the diverse phenotypic variation seen in both yeast and prion diseases[20]. Several mammalian pathological proteins have also been shown to adopt distinct self-propagating aggregates or "strains" with different structures, which are presumably linked to the phenotype diversities of degenerative diseases[16,42–44]. One such example is Tau protein, deposition of whose pathological forms results into Tauopathies, which includes Alzheimer's disease, Fronto-Temporal Dementia (FTD). Several studies support the "tau strain

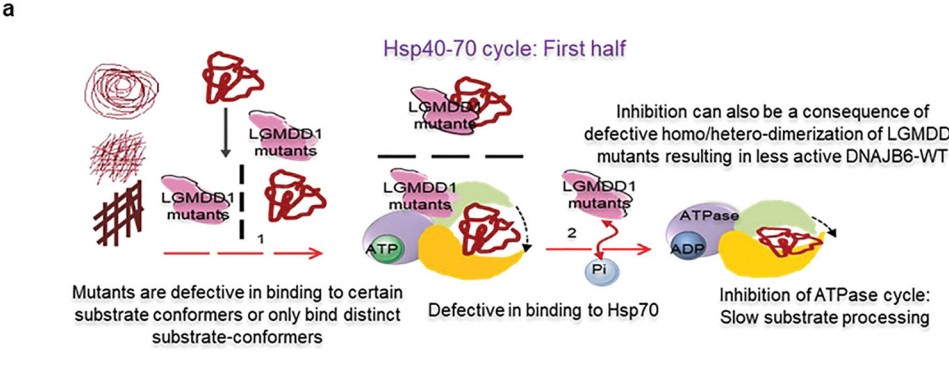

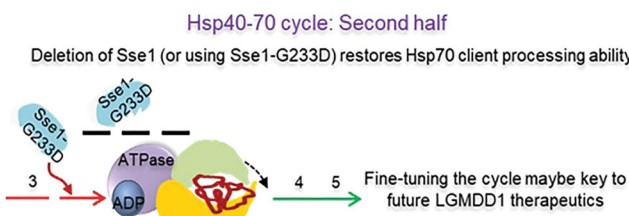

**Fig. 8 | Schematic diagram depicting possible mechanism of LGMDD1 mutants and their effects on the Hsp70 ATPase cycle, as well as proposed therapeutic intervention. a** Cartoon depicting possible mechanistic insights as to how LGMDD1 G/F domain mutants delay or inhibit the Hsp70 ATPase cycle. The dashed red arrows indicate inhibition. Mutants were inefficient in terms of both homo and hetero-dimerization and that may be related to their reduced ability to assist Hsp70 in protein folding. In the first half of the Hsp70 ATPase cycle, LGMDD1 mutants were incompetent in their ability to bind specific substrate conformers and Hsp70 (black dashed arrows), which delays the downstream processing of the substrate through Hsp70-ATPase cycle. This inhibition could negatively impact Hsp70-mediated ATP-hydrolysis. **b** Possible therapeutic route. In the second half of the cycle, modulation of NEF inhibit the binding and exchange of nucleotide, which delay the downstream process of substrate release in the cycle. However, this delay led to positive consequences in terms of yeast prion propagation (green arrow represents restoration of normal function). Thus, altering the balance between the two halves of the Hsp70-ATPase cycle may provide one route for therapeutic intervention for these types of diseases.

hypothesis", which proposes that different aggregated tau conformers (distinct strains) have distinct pathology-initiating capacities because they interact with endogenous tau differently[45,46]. In yeast, the interaction between the prion protein Rnq1 and the Hsp40 Sis1 is required for [*RNQ*+] propagation[29]. Therefore, we utilized this system to ask whether the LGMDD1 mutants impacted the [*RNQ*+] prion strains that form de novo. Strikingly, we found a change in [*RNQ*+] prion strain formation when Rnq1 fibers were formed in the presence of the LGMDD1 mutants as compared to wild type Sis1. These changes were a direct consequence of Hsp40 interaction alone and may be a consequence of DNAJ proteins ability to act as "*holdases*"[47,48]. "*Holdases*" are chaperones that do not use ATP and simply protect their client protein from aggregation[48]. Unlike "*holdases*", "*foldases*" (like Hsp70) accelerate the transition of non-native conformations towards native states in an ATP-dependent manner. Interestingly, the proteostasis network relies on a constant interplay between these two kinds of chaperones. Putative "*holdase*" activity is not shown in the Hsp40/70-cycle and may well be altered with the LGMDD1 mutants (Supplementary Fig. 10).

As a major role for Hsp40s is stimulating Hsp70s, another important aspect of the LGMDD1 mutants might be a change in the productive interaction with Hsp70[25]. Indeed, we found that these LGMDD1 mutants were defective in binding to Ssa1 (Hsp70) as well as to substrates (Rnq1 and luciferase). It had been shown previously that there was no difference between Sis1-WT and the Sis1-G/F domain knockout induced Ssa1 ATPase activity in absence of substrate[25]. Interestingly, however, the LGMDD1 mutants showed a reduction in the stimulation of ATPase activity of Hsp70. Moreover, in the presence of substrate, the change in ATPase activity was client-conformer specific. Additionally, LGMDD1 mutants were defective in refolding heat-

denatured luciferase (similar to previous findings with Sis1ΔG/F[25]), indicating a more global defect in substrate remodeling. These findings fit with recent data that implicate the G/F-rich region of DNAJB1 in an autoinhibitory mechanism that regulates the major class B J-domain proteins (JDPs)[49]. This inhibition can be released with second site mutations (E50A, or F102A, and or ΔH5) in DNAJB1[49]. A similar mechanism has been suggested for both DNAJB6 and DNAJB8 proteins[50,51]. Our data suggest that the LGMDD1 G/F domain mutants are not simply releasing the auto-inhibitory mechanism, however, as they show reduced binding to Hsp70 in the absence of client, indicating that the complex (Sis1-substrate-Hsp70) formed with Hsp70 is indeed driven by the Sis1-substrate interaction.

The function and interaction of the various Hsp40 domains have been studied extensively[13,52]. The efficiency of Hsp70 ATPase activity is heavily dependent on the proper functioning of the J and G/F domains of Hsp40[52,53]. Moreover, Hsp40 (Sis1) functions as a dimeric protein[32]. Interestingly, we found that LGMDD1 mutants show a diminished ability to form homodimers and heterodimers (with Sis1-WT). However, DNAJB6 has not been described as a dimer (and is instead polydispersed) and has a CTD that is distinct from that of Sis1-WT or DNAJB1. Thus, the decreased dimerization efficiency observed with the homologous mutants in Sis1 just provides further insight into other possible changes with DNAJB6 G/F domain mutants.

Previously, it was suggested that the LGMDD1-causing mutations exert a deleterious dominant negative effect on the wild-type protein[3]. An excess of mutant (DNAJB6-F93L) to wild-type mRNA induced lethality in embryos, while an excess of wild-type to mutant mRNA gave rise to progressively increased rescue[3]. Consistent with this, we found that titrating Sis1-WT with LGMDD1 mutants (F106L and F115I) decreased Ssa1 ATPase activity.

Interaction of the Hsp70/40 machinery with a misfolded client is not sufficient to promote re-folding. The regulated cycle of client release and the potential for re-engagement is important, and this is dependent on nucleotide exchange. Indeed, changes in the availability or function of nucleotide exchange factors (NEFs) alone change client processing. Alterations in the NEF Sse1 were shown to alter yeast prion propagation in a strain-dependent manner[54]. Sse1 has been proposed to have multiple functions and can act as a disaggregase itself[55,56]. Our data suggest that the effect of the LGMDD1 mutants on the propagation of the [RNQ+] strain can be rescued by the deletion of Sse1 (Hsp110). Notably, although the deletion of Sse1 alone showed partial improvement in [RNQ+] prion propagation in yeast cells, it did not show any difference in the refolding of luciferase, again perhaps indicating substrate specificity. This indicates that for some clients, these mutants might be rescued by modulating the time of interaction with Hsp70/40.

We hypothesize that the observed defects in the LGMDD1 mutants result in cellular phenotypes that are client-conformer specific. The Hsp70/DNAJ ATPase cycle is a process partitioned into two interconnected events; DNAJ is vital in the first half whereas NEFs play a significant role in the second half. Our data suggest that there are numerous defects associated with DNAJB6 (Sis1) mutants which result in either inhibition or delay in client processing in the first part of the Hsp70-ATPase cycle. Hsp70 has been shown to suppress proper substrate folding if it is not allowed to cycle off its client protein in various contexts[57,58]. Henceforth a longer interaction of LGMDD1 mutants with Hsp70 might lead to broader disruption of Hsp70-dependent processes, as this could titrate Hsp70 away from other clients[59]. Our data suggest that inhibiting the second half of the ATPase cycle, can have positive consequences on client processing. Interestingly, previous data suggest that the optimal NEF activity for protein disaggregation occurs at a reduced ratio of NEF:Hsp70 (1:10)[60–62], and perhaps the deletion of Sse1 recapitulates such reduction in NEF activity in some manner (such as replacing the optimal NEF with another, such as Fes1). Moreover, the armadillo-type NEFs (budding yeast Fes1 and its human homolog HspBP1) employ flexible N-terminal release domains (RDs) with substrate-mimicking properties to ensure the efficient release of persistent substrates from Hsp70[63]. This is plausible due to the fact that NEFs perform dual functions: accelerating nucleotide exchange and securing Hsp70-liberated substrates. Of note, the high selectivity of exchange factors for their Hsp70 partner contributes to the functional heterogeneity of Hsp70 chaperone system[64]. These results indicate that fine-tuning of the two halves of the Hsp70 ATPase cycle involving LGMDD1 mutants and NEFs during the processing of its client proteins is critical (Fig. 8b). As such, we suggest that NEF inhibitors could provide a possible therapeutic strategy for the treatment of LGMDD1. There are certain aspects of DNAJB6 activity which are imperfectly modeled by Sis1. Nevertheless, in the absence of a clear mammalian substrate for DNAJB6, these findings with its closest yeast homolog Sis1 and the impact of the LGMDD1 mutants on client processing provide crucial mechanistic insight to begin to explore therapeutic routes for this myopathy.

## Method

### Cloning, expression and purification of recombinant proteins

Sis1-WT, Sis1-mutants (F106L/N108L/D110Δ/F115I/ΔDD) and Ssa1-WT were cloned into pPROEx-Htb vector obtained from Addgene. The plasmid encodes a hexa-His-tag, a TEV cleavage site, and the respective cloned gene for expression. All Sis1 mutants were generated using the Quick Change Mutagenesis Kit (Agilent Technologies #200517). Primer sequences were generated using Agilent's online primer design program. Mutagenesis was confirmed by sequencing the entire coding region of Sis1. Sis1-WT and Sis1-mutants were expressed at 16 °C, whereas Ssa1-WT was expressed at 18 °C to increase the fraction of soluble protein. All purification steps were carried out at 4 °C. Protein purity was more than 99% as determined by SDS/PAGE and Coomassie staining. Final protein concentration was estimated by Bradford assay, using bovine serum albumin as the standard. Following purification, all the proteins were frozen on liquid nitrogen and stored at −80 °C till further use. Sis1-WT and mutants were purified using standard protocol with some modifications. Briefly, these were purified from Escherichia coli strain Lemo 21(DE3) (New England Biolabs #C2528H) grown in 2X YT medium at 30 °C until $OD_{600} = 0.6-0.8$. The cultures were induced with 0.5 mM IPTG and grown overnight at 16 °C. Cells were harvested and lysed in buffer A (50 mM Sodium phosphate buffer (pH 7.4), 300 mM NaCl, 5 mM $MgCl_2$, 10 mM Immidazole, 0.1% Igepal, 0.01 M TCEP (tris(2-carboxyethyl)phosphine) (Thermo-Fisher Scientific #20491), protease inhibitor cocktail (EDTA-free) (Thermo-Fisher Scientific #A32965) and a pinch of DNase I (Millipore Sigma #DN25). Cell debris was cleared by centrifugation (20,000 × g) and the supernatant loaded on cobalt-based Talon metal affinity resin (Thermo-Fisher Scientific #89964). After washing, proteins were eluted as gradient fractions with buffer A containing increasing concentrations of imidazole (150–400 mM) (Millipore Sigma #I0125). Purified proteins were incubated with His-TEV (purified in the lab) protease at 30° for 1 h. The samples were extensively dialyzed at 4 °C and again passed through Talon metal affinity resin to remove the cleaved His tag and His-TEV protease. The pure proteins were concentrated and stored at −80 °C. Similarly, Ssa1-WT protein was also purified using an established procedure[65]. Briefly, protein was purified from Escherichia coli strain Rosetta 2(DE3) (Novagen, EMD-millipore #71397-3) grown in LB medium with 300 mM NaCl at 30 °C until $OD_{600} = 0.6-0.8$. The culture was induced with 0.5 mM IPTG and grown overnight at 18 °C. Cells were harvested and lysed in buffer A (20 mM Hepes, 150 mM NaCl, 20 mM $MgCl_2$, 20 mM KCl, protease inhibitor cocktail (EDTA-free) using lysozyme (Millipore Sigma #L6873). The rest of the protocol was similar to that of Sis1-WT and mutants, with only exception being that Ssa1-WT was eluted with buffer A containing 250 mM imidazole. Rnq1-WT full-length protein was purified exactly as described in previous publication from our lab[26].

### Binding assays

**Substrate-binding ELISA assays.** This was performed as described earlier[25] with some modifications. Two different substrate proteins-Rnq1 and firefly luciferase (Millipore Sigma #SRE0045) were denatured for 1 h at 25 °C in a buffer comprising of 3 M guanidine HCl, 25 mM HEPES (pH 7.5), 50 mM KCl, 5 mM $MgCl_2$, and 5 mM DTT. Following denaturation, substrates were diluted in 0.1 M $NaHCO_3$ and bound to microtiter plate (CoStar 3590 EIA plates, Corning #CLS3590) at a concentration of 0.4 μg/well for Rnq1 and 0.1 μg/well for luciferase, respectively. Unbound substrate was removed by washing with phosphate buffered saline (PBS). Unreacted sites were blocked with 0.2 M glycine (100 μl/well) for 30 min at 24 °C, followed by washing with PBS-T (PBS containing 0.05% Tween 20). Non-specific binding was eliminated by blocking with 0.5% fatty-acid-free bovine serum albumin (BSA) (Millipore Sigma #A6003) in PBS for 6 h. The wells were subsequently washed with PBS-T. Sis1-WT and Sis1-mutants were serially diluted in PBS-T (substituted with 0.5% BSA) and incubated with substrate for overnight at 24 °C. After extensive washing with PBS-T, rabbit anti-Sis1 antibody (CosmoBio #COP-080051) at a dilution of (1:15000) was added and incubated for 2 h at 24 °C. This was followed by further washings and the addition of donkey-anti-rabbit HRP-conjugated (1:4000) (Millipore Sigma #AP182P) as secondary antibody. The amount of Sis1 retained was determined by developing a reaction using tetramethyl benzidine/$H_2O_2$ (TMB peroxidase EIA substrate) kit (Bio-Rad #1721068)). The color was measured at 450 nm (SpectraMax M2e fluorimeter microplate reader) after terminating the reaction with 0.02 N $H_2SO_4$.

**Ssa1-binding assays.** Ssa1 (200 nM) was immobilized in microtiter plate wells and dilutions of purified Sis1-WT and Sis1-mutants were incubated with it. Bound Sis1 was detected as described above.

**Substrate bound Sis1 combined with Ssa1 binding assays.** Denatured substrates (Rnq1 and luciferase) at a concentration of 0.4 μg/well and 0.1 ug/well, respectively, were mixed with Sis1-WT and Sis-mutant proteins (5 nM) and incubated for 1 h at 24 °C, prior to being adsorbed in the microtiter well plates. Following the steps of incubations, washings and blocking, serially diluted Ssa1-WT was added to each well. Subsequently, the wells were probed with mouse-Hsp70 (Ssa1) antibody (1:2000) (Abcam #ab-5439), followed by goat anti-mouse IgG [H + L] HRP conjugated secondary antibody (1:4000) (Thermo-Fisher Scientific #62-6520). The detection method used was similar to that described above for detecting Sis1.

**Homo/Hetero dimeric nature of Sis1 determining binding assays.** Serially diluted His-tagged cleaved Sis1-WT and Sis1-mutants were adsorbed in the microtiter well plates. Following washings, serially diluted uncleaved His-tagged Sis1-WT and Sis1-mutants were added to the wells in the following combinations [(Cleaved mutants + Uncleaved mutants = Homodimer); (Cleaved mutants + Uncleaved Sis1-WT = Heterodimer) along with appropriate controls for the assay. This was followed by washings, blocking and addition of mouse anti-His antibody (1:5000) (Novagen #37-2900). Rabbit anti-mouse HRP conjugated antiserum (1:4000) (Millipore Sigma #AP160P) was used as secondary antibody. The detection method was similar to that described above.

**Amyloid fiber formation and thioflavin T kinetics**
Purified Rnq1 was resuspended in 7 M guanidine hydrochloride (Millipore Sigma #G3272) and the protein concentration was determined. Rnq1 fibers were formed at 18, 25, and 37 °C with a starting monomer concentration of 8 μm in Fiber-formation buffer (FFB) (50 mM KPO$_4$, 2 M Urea, 150 mM NaCl, pH 6). For the seeded kinetics experiments, the fibers were seeded using 5% (w/w) seed. The fiber formation and kinetics assays were performed in the presence of Thioflavin T dye (Millipore Sigma #T3516) and acid-washed glass beads (Millipore Sigma #G8772) for agitation as described earlier[26]. Kinetic assays of fiber formation were done in a SpectraMax M2e fluorimeter microplate reader. The change in Thioflavin-T fluorescence over time was measured using an excitation wavelength of 450 nm and emission wavelength of 481 nm.

**Colorimetric determination of ATPase activity**
The ATPase assay was performed as described before[66] with some modifications. Briefly, the ATPase reagent was made by combining 0.081% W/V Malachite Green (#M6880) with 2.3% W/V poly-vinyl alcohol (#363138), 5.7% W/V ammonium heptamolybdate (#09878) in 6 M HCl (#258148), and water in 2:1:1:2 ratios (all purchased from Millipore Sigma with no further purification). This ATPase reagent was freshly prepared every day and was left standing for 2 h to get a stable green/golden solution, which was filtered through 0.45 μm syringe filters (Millipore Sigma #SLHA033SS) before use. ATPase activity in the absence of any client protein was tested by incubating Sis1-WT/mutants, Ssa1-WT in the ratio of (0.05:1.0 μM) with 1 mM ATP, in assay buffer (0.02% Triton X-100, 40 mM Tris–HCl, 175 mM NaCl, and 5 mM MgCl$_2$, pH 7.5) at 37 °C for different time-intervals as indicated in the figure legends. At the end of incubation 25 μL of the reaction was added to a well in a 96 well plate, followed by 800 μL of the ATPase reagent and 100 μL of 34% sodium citrate to halt any further ATP hydrolysis. The mixture was allowed to incubate for 30 min at 24 °C before absorbance at 620 nm was measured using a SpectraMax M2e fluorimeter microplate reader. A sample of ATP alone in buffer was treated exactly the same and was subtracted from the sample absorbance to account for intrinsic ATP hydrolysis. To account for variability in measurements a phosphate standard curve (using potassium phosphate) was created for each day of measurements. For ATPase activity in presence of client proteins, the same procedure was followed with only exception being the addition of client protein Rnq1 monomer (25 μM) and Rnq1 seeds (10% of which is used in final reaction) formed at three different temperatures (18, 25, and 37 °C) with the chaperones and ATP for incubation.

**Luciferase refolding assay**
Heat denatured refolding of luciferase was performed as previously described[67]. Briefly, Ssa1 (2 μM) were incubated in refolding buffer (50 mM Tris pH 7.4, 150 mM KCl, 5 mM MgCl$_2$) supplemented with 1 mM ATP and an ATP regenerating system (10 mM phosphocreatine, 100 mg/ml phosphocreatine kinase) for 15 min at room temperature. Next, luciferase (25 nM) was added and incubated for a further 10 min. Then Sis1-WT/mutants (0.05 μM) was added and the mixture was heat shocked at 44 °C for 20 min. The reactions were then immediately moved to room temperature. Finally, 25 μL aliquots of the refolding reaction were then taken and added to 50 μL of luciferase assay reagent (Promega Corporation #E1501). At various time points, activity was then measured with a GloMax Luminometer (Promega Corporation).

**Titration assays**
All the assays were performed by titrating the concentration of Sis1-mutants (F106L and F115I) with Sis1-WT (concentration been kept constant).

**Yeast strains, plasmids and Transformation**
The yeast strains used in this study are derived from *Saccharomyces cerevisiae* 74-D694 (*ade1-14 his3-Δ200 leu2-3, 112 trp1-289 ura 3-52*). Yeast cells were grown and manipulated using standard techniques[68]. As indicated, cells were grown in rich media YPD (1% yeast extract, 2% peptone, 2% dextrose) or in synthetic defined (SD) media (0.67% yeast nitrogen base without amino acids, 2% dextrose) lacking specific nutrients to select for appropriate plasmids. Wild-type (WT) yeast harboring the s. d. medium [*RNQ*+] variant and the [*rnq*-] control strain were kindly provided by Dr. Susan Liebman (University of Nevada, Reno, Nevada, USA)[69]. Construction of Δ*sis1* [*rnq*-] and s. d. medium [*RNQ*+] yeast strains were described previously[12]. Δ*sse1* was created using a plasmid-based *sse1Δ::LEU2* disruption construct (a kind gift from Dr. Kevin Morano) that was digested with SacII/PstI and transformed into *sis1Δ* cells propagating s.d. medium [*RNQ*+] and expressing Sis1 from a *URA3*-marked plasmid. All other delta strains in s. d. medium [*RNQ*+] background were created using the standard protocol. Medium containing 1 mg/mL 5-fluoroorotic acid (5-FOA) that selects against cells maintaining *URA3*- marked plasmids was used to replace WT Sis1 with the mutant constructs using the plasmid shuffle technique. Plasmid transformations were done using polyethylene-glycol/lithium-acetate (PEG/LioAC) technique, and the cells were selected using SD-trp/his plates.

Plasmid pRS316-*Sis1* was kindly provided by Dr. Elizabeth Craig (University of Wisconsin, Madison, Madison, Wisconsin, USA)[22]. Plasmid pRS414-*Sse1-G233D* was a kind gift from Dr. Kevin Morano (McGovern Medical School, UT Health, Houston, Texas, USA)[37]. Construction of pRS314-*Sis1* and LGMDD1 mutants were described previously. Using the standard molecular techniques we constructed p413TEF-*Sse1-G233D*. Plasmid pRS316-*GPD-Lux* was a kind gift from Dr. Bernd Bukau (Center for Molecular Biology of Heidelberg University, Heidelberg, Germany)[39].

**Protein fiber transformation for phenotypic analysis**
Transformation of Rnq1 fibers into a [*rnq*−] 74-D694 (*ade1-14, ura3-52, leu2-3,112, trp1-289, his3-2OO, sup35::RRP*) yeast strain in the presence and absence of Sis1-WT/ mutants was conducted as described[70]. The

resulting colonies formed after infecting fibers formed in vitro in the presence and absence of chaperones were replica plated onto rich medium (YPD) plates to assay for colony color. Colonies that appeared to have acquired the prion state by nonsense suppression were picked and spotted on YPD, YPD containing 3 mM GdnHCl, and SD-Ade for phenotypic analyses.

### Protein analysis

Yeast samples were lysed with glass beads in buffer (100 mM Tris-HCl pH 7.5, 200 mM NaCl, 1 mM EDTA, 5% glycerol, 0.5 mM DTT, 3 mM PMSF, 50 mM N-ethylmaleimide (NEM), complete protease inhibitor from Roche) and pre-cleared at 6000 rpm for 15 s. Protein concentration of cells lysates was then normalized. For well-trap assays, samples were incubated at room temperature or 100 °C in sample buffer (200 mM Tris-HCl pH 6.8, 4% SDS, 0.4% bromophenol blue, 40% glycerol), then analyzed by SDS-PAGE and western blot using an αRnq1 antibody (1:2500) and mouse anti-rabbit HRP conjugated secondary antibody (1:10000). Boiled gel assays were performed as described previously[15]. Briefly, yeast cells were lysed with glass beads in buffer (25 mM Tris-HCl pH 7.5, 100 mM NaCl, 1 mM EDTA, protease inhibitors) and pre-cleared at 6000 rpm for 1 min at 4 °C. Protein concentration of cell lysates was normalized using a Bradford assay and mixed with SDS-PAGE sample buffer (200 mM Tris-HCl pH 6.8, 4% SDS, 0.4% bromophenol blue, 40% glycerol). Samples remained un-boiled and were loaded on a 12% polyacrylamide gel and run under constant current of 110 V until the dye front migrated halfway through the resolving gel. The current was then stopped, and the gel in glass plates was sealed in plastic and boiled upright for 15 min in a 95–100 °C water bath. After boiling, gels were removed from the plastic cover and were reinserted in the SDS-PAGE apparatus, where voltage was re-applied until the dye migrated to the bottom of the gel. SDS-PAGE was followed by standard western blotting with αRnq1 antibody.

### Negative staining transmission electron microscopy

Rnq1 fibers were generated as described above. Negative staining of Rnq1 samples was performed by depositing 8 μL of sample and incubating for one minute on carbon coated 200 mesh copper grids (01840-F, Ted Pella, Redding, CA), held by forceps carbon side up, which had been freshly glow discharged for 30 s in a Solarus 950 plasma cleaner (Gatan, Pleasanton, CA). Post-incubation, each grid was washed five times in separate ultrapure water droplets and subsequently stained with 0.75% uranyl formate for 2 min. Excess uranyl formate was blotted off using filter paper (Whatman No.2, Fisher Scientific, Hampton, NH) and subsequently air dried. Sample grids were imaged using a JEOL JEM-1400 Plus Transmission Electron Microscope operating at an accelerating voltage of 120 kV equipped with an NanoSprint15 MKII sCMOS camera (AMT Imaging, Woburn, MA). Images were acquired using a total exposure time of 5 s containing ten 500 ms drift frames which were subsequently aligned and averaged using the AMT ImageCapture Engine acquisition software (version V701) at nominal magnifications ranging between 20,000–50,000×.

### Statistical analysis

All data are analyzed using GraphPad Prism (v8, https://www.graphpad.com/scientific-software/prism/), and Microsoft Excel 2016 (https://www.https://www.microsoft.com/en-us/microsoft-365/microsoft-office) software. Error bars represent standard error mean from at least three experiments. Significance was determined for two-sample comparisons using the unpaired $t$-test function with a threshold of two-tailed $p$ values less than 0.05 for *, 0.01 for **, and 0.001 for ***.

### Reporting summary

Further information on research design is available in the Nature Research Reporting Summary linked to this article.

## Data availability

The data that support this study are available from the corresponding author upon reasonable request. Source data are provided with this paper.

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

## Acknowledgements

We are thankful to S. Liebman, J. Weissman, K. Morano, and E. Craig for plasmids and strains. We also thank Kevin Stein for initial experiments with Sse1. We are thankful to Patrick McConnell and Kevin M. Kaltenbronn for helping with protein purification and luciferase assay, respectively. We are thankful to members of True lab and Weihl lab for helpful discussions and comments on the manuscript. This work was supported by: National Institutes of Health Grant R01AR068797 to H.L.T. and C.C.W. The funders had no role in study design, data collection and analysis, decision to publish, or preparation of the manuscript.

## Author contributions

A.K.B., C.C.W., and H.L.T. conceived and designed the experiments and analyzed data. A.K.B. performed all the experiments and wrote the initial draft. M.J.R. and J.A.J.F. collected the negative-staining TEM images. J.A.D. performed the site-directed mutagenesis. A.K.B., C.C.W., and H.L.T. reviewed and edited the manuscript. All authors have provided editorial input.

## Competing interests

The authors declare no competing interests.
