## [Peer Review File · Nature Communications]

Disease-associated mutations within the yeast DNAJB6 homolog Sis1 slow conformer-specific substrate processing and can be corrected by the modulation of nucleotide exchange factorsReviewers' Comments:

Reviewer #1:

Remarks to the Author:

This manuscript by Bhadra et al describes an interesting investigation - how some pathogenic mutations of human DNAJB6 that cause dominant myopathy might alter its binding ability to substrates and its interactions to co-chaperones by using the budding yeast *S. cerevisiae* as a testing system. *S. cerevisiae* contains several yeast prions, including [RNQ+] which is a substrate of Sis1 – an essential J-protein homologous to DNAJB6. A set of myopathy mutations that occurred in DNAJB6 were generated in Sis1 and the authors found that those mutants have compromised functions in client protein binding, denatured protein refolding, and co-chaperone interactions. These findings provide a mechanistic insight into the pathogenic effect of these mutations on Hsp70/40 ATPase process. [RNQ+] can exist in multiple conformers – prion strains and interestingly, the authors found that different Sis1-myopathy mutants can have a very different effect on de novo formation of different [RNQ+] strains, in other word, prion conformer-specific. Although how much the knowledge learned from here can be transformed into the mammalian skeletal muscles remains to be tested, their finding is interesting and potentially important. It may provide new approach to explore potential therapeutics of myopathy along this line of research. I recommend it be accepted by Nature Communication after the following concerns are addressed:

- 1) The manuscript would be strengthened if the authors show the quality of the Sis-WT and Sis1-mutants (SDS-PAGE) that were used in this study.
- 2) Although the conclusion that the impact of Sis1 on Rnq1 fiber formation is mainly at the nucleation step might be correct, the authors should present their data of Fig S1A, B, and C in a clearer format. I suggest use a same scale of Y-axis for all temperatures; and/or calculate the rate of Rnq1 fiber elongation (in the presence of seeded Rnq1 fiber) for each mutant in each temperature and compare them in a table. The current form of graphs is too busy to give a clear picture on what is going on.
- 3) Line 219 to 220: the sentence “Once the mutants bind substrate, however, they are able to stimulate-Hsp70 activity (Fig 3D)” does not make sense. Need to clarify what evidence to allow drawing such conclusion.
- 4) Supp. Fig. 1D – the quality of the figure is questionable – poor resolution and overexposure. I suggest that the authors either provide a better image to show the differences between the samples or remove this figure.
- 5) Experiments presented in Fig 6 and supp Fig 5 have some designing problems. The amount of Sis1-WT is different in each examined mixture. One should expect to see Ssa1 ATPase activity being reduced when the concentration of the Sis1-WT is decreased even though the total concentration of Sis1 plus mutant Sis1 remains same as both Sis1 mutants were shown to have a significantly less activity. To make the point that Sis1-WT dimerization is inhibited by a mutant Sis1, one should keep the concentration of Sis1-WT same across the board while increasing the mutant Sis1 concentration. In this design, you also need to include a set of controls: Sis1-WT alone, mutant Sis1 alone for each concentration that will be added to the mixture. Whether a Sis1 mutant inhibits Sis1 WT dimerization can be seen by comparing these treatments.

Reviewer #2:

Remarks to the Author:

In the current MS, the authors have made several Sis-1 mutations that match with some of the DNAJB6 mutations causing Limb-Girdle Muscular Dystrophy (LGMD) and tested their impact on Sis-1 function using both yeast and in vitro data with purified proteins. To me, the take-home message is

that the Sis-1 mutants show (partial) loss of function in their “holdase” function to keep substrates in a non-aggregated form as well as in their ability to interact with Hsp70 (Ssa1) to stimulate its ATPase activity.

My conclusion from the data is further that whereas the loss of holdase function depends much on the type of the G/F mutant used (different in extent for different substrates), the defect on stimulation the ATPase of Ssa1 seems rather similar for the different mutants. The latter, however, is hard to extract from the data as most experiments focus on co-stimulation of substrates plus Sis-1 (mutant) that are obviously confounded by the fact that Sis-1 mutants differently interact with those substrates. So, to confirm this would require more in-depth analysis of the effects of (different concentrations) of Sis-1 and mutants alone on the ATPase of Ssa1.

A main concern is further that whereas Sis-1 shares many functional features with DNAJB6 (for which mutation cause LGMD), it is yet unclear that it is a true homologue. Clearly, Sis-1 also share functional features with human DNAJB1. In fact, features of supporting Hsp70-dependent refolding of heat denatured luciferase (one of the substrates used here) has been demonstrated for DNAJB1 but not DNAJB6. Moreover, AlphaFold predictions, show that Sis-1 shares structurally features of both. So, even though the current data with equivalent LGMD mutation in Sis-1 are very informative, some cautionary remarks must be added.

Also, the paper suffers from severe real inaccuracies, jargon and sometimes lack of information that make it hard to digest. All figures are rather sloppy: it is a puzzle to find out with symbols belong to which mutants. All labelling and coloring should be improved, and I suggest using clearer and fixed colors/symbols for the wildtype and mutant Sis-1 variant throughout all figures.

However, the paper does contain very interesting data that, if presented better, more cautiously discussed and maybe supplemented by some additional data at places would very valuable to the chaperone and protein homeostasis and disease community. The findings that both substrate binding and Hsp70 interactions are each (slightly) impaired in these mutant also helps to elucidate the yet unclear role of the G/F rich region in class A and Class B J-domain proteins and help to understand the functional impact related to the recent findings that this region may actually be involved in an (subtle) autoinhibition of these classes of chaperones.

Specific comments:

- Figure 1 shows that the G/F mutants of Sis-1 have reduced capacities to delay amyloid formation of Rnq1, the F115L mutant showing the largest LOF.

It would be helpful to quantify the data (e.g. time to 50% of ThT RFU) and order the mutants (in my view, the order looks like: F115L < D110delta < F106L < N108L < wt).

What does confuse me a bit is the EM data: these show complete absence of fibers in the presence of F115L whilst fibers were formed in the presents of wildtype Sis1 (albeit of different structure). I do not know how this can be matched with the ThT data.

The mutants all also have slight defects in assisting Hsp70-dependent luciferase, but the extent here seems similar for all mutants. Actually, these data should include data points immediately after HS to test if the major effect was not on the extent of luciferase inactivation only, as it seems that the refolding kinetics are not much affected. Also note, that DNAJB6 was found to be unable to assist refolding, whereas DNAJB1 did (illustrating my general remark that some caution must be considered to directly translate findings with Sis-1 to DNAJB6). Finally the order of loss of function for this endpoint should be somehow provided (as far as I can see, it now ranks as all mutants show a similar defect: F115L = D110delta = F106L = N108L < wt)

- Figure 2: The labelling is totally unclear, especially panel B: what samples are shown here? Again, try to order the mutants in terms of deficiency (in my view: F115L < F106L < wt). Why not also include the 2 other mutants to link this better to the order of deficiency seen in the ThT assay? Please also note that (to my knowledge) a role for the G/F region in substrate interactions has only been suggested for Sis-1 and DNAJB1 but for DNAJB6 (as yet).

- Figure 3 + Fig S3 are intriguing as they suggest that the mutants have reduced capacity to stimulate the ATPase of Ssa1. Whilst the authors state that this effect is only seen when substrates are added, my impression of the data in Fig S3 are that this is not incorrect. And in fact, it would be highly important to better analyse this. I do know that combined JDP and substrates have larger effects on Hsp70 ATPases than single components only. To me the data rather suggest that the different RNQ1 substrates are different in their extent of Hsp70 ATPase stimulation (such data should be provided) and that hence this adds up differently with the effects of the Sis-1 mutants (that all may share the same extent of a minor but functionally relevant defect). Also, given that the Sis-1 variants have different interactions with the substrates make their combine effects on the Sis-1 ATPase hard to interpret.

Moreover, the raw data as presented are also not easy to evaluate quantitatively: the authors should determine rate parameters to better allow comparison of the various conditions.

Further, if I am correct, fibers were formed and then added together with Sis-1. As their data in Figure 1/S1 showed that Sis-1 does not work on seeding aggregation, this may not directly be relevant to what is seen in the ThT.

- Figure 4: This figure now directly analyses the direct binding of the Sis-1 variants to the 2 substrate investigated in Figure 1/2 (in fact, therefore this figure should probably be best presented directly after these functional data). Again, the data should be analysed to provide a ranking of binding efficiency. As far as I can now see this ranking for Rnq1 (D110delta < F106L < N108L < F115L < wt) and Luciferase (D110delta < F106L < F115L < N108L < wt) do not match with the functional data. How do the authors explain this? What should we learn from this?

In this figure binding to ssa1 is also include, which should be more directly compared to the ATPase data in Figure 3 and should also include data on the binding of the Sis-1 without substrate. What is striking to me here that, at least in the presence of substrates all Sis-1 mutants bind LESS to Hsp70, with almost no difference between them. First, how do we need to interpret this with regards to the data in figure 3? Second, would one not expect that, if the G/F mutants would release the auto-inhibitory loop on Hsp70 accessibility that it should be more to Ssa1? This is also why data regarding binding between Ssa1 and Sis-1 mutants without the substrates seem essential here.

- Figure 5: This data should suggest that the mutant of Sis-1 are impaired in forming a dimer. First, I do not know precisely what exactly is expressed and what it implies: does the mutants still form some dimers? The data should actually include a positive control (CTD deletion mutant lacking the dimerization dimer). Also, what do these data mean in terms of the functionalities described in figure 1-4? Also for that a CTD deletion mutant may be required for comparison in these figure 1-4. Finally, note that DNAJB6 has not being described as a dimer (rather as polydispersed) and has a rather different CTD than Sisi-1 or DNAJB1 and hence the relevance of this figure for understanding the LMDG-mutants is a bit unclear to me.

- Figure 6: It is repeatedly written in the MS, in particular in relation to this figure that the mutants are dominant in nature. Whereas it is clear that DNAJB6-related LGMD is indeed dominant, the data on the mutants so far show partial loss of function, which could mean the disease may also be due to haploinsufficiency. The data in figure 6 also do not show, in my view, evidence for dominant negative effects, but only support partial loss of function. To demonstrate dominant-negative effects, one should not used different ratios of wildtype and mutants at a fixed total concentration, but rather use a fixed concentration of the wildtype protein and add increasing concentration of the mutants: if would

result in a bell shaped curve of activity this would suggest dominant negative effects. So, as is stands, I see very little added value of figure 6.

- Figure 7: This figure (and its accompanying Fig S7) was presented in a for me very confusing manner. First of all, the data factually show Rnq1 aggregation and not directly prion propagation, which -I know- is linked and prion jargon, but yet confusing. The experiments should be explained better and also the figure is totally confusing (panel A, upper and lower panel are not even explained). What the data show (as far as I could follow....) is that depletion of Sse1 affected prion handling, but this is true for both wildtype and mutants, implying that increased dwell time on Hsp70 may be beneficial for this substrate. Too me, it is also a total overinterpretation of the data to state that Hsp70 activity inhibition provides a potential mechanism for therapeutic intervention for LMGD (line 272-2740) as it seems from the data that both substrate affinity of the Sis-1 mutants as well as interactions with Hsp70 are impaired. So, the machinery as a whole seems mistuned. I could not decipher the data in Figure 7B, let alone the related conclusion.

- Discussion: The presented model is very confusing and several statements in the text of the discussion seem to be imprecise: E.g.,
 - There are no data showing "client processing" is affected by LGMDD1 G/F domain mutants. (lines 311-314). Indeed, as stated later, their substrate "holdase" capacity seems affected (lines 339-340)
 - also, the mutants do not "inhibit" Hsp70 ATPase (lines 315-316); they just stimulate and do so less well than wildtype!
 - there was no difference in the ATPase activity between wildtype and mutants (lines 344-345): Sis-1 has no ATPase activity, it stimulates that of Ssa1, and as stated above this activity IS reduced as is clearly evident from Figure S3B. This requires additional investigations though (as stated above)
 - Indeed, the G/F-rich regions of DNAJB1 have been implicated in an autoinhibitory mechanism that regulates its interactions with Hsp70. Note and cite, that this has now also been suggested to be the case for DNAJB6/8 (as this is where the LGMD mutations are relevant for).
 - Lines 369-373: As stated above: I disagree that the current data are indicative of a dominant negative effect.

Reviewer #3:

Remarks to the Author:

While a great deal of molecular detail is available regarding how molecular chaperones recognize and interact with substrate/client proteins, especially in vitro, how altered functions lead to disease is far less well understood. The authors of the manuscript under consideration previously established that mapped patient mutations in the locus DNAJB6, encoding a cytosolic Hsp40 chaperone family member, are linked to a myopathy termed Limb-Girdle Muscular Dystrophy Type D1. However, neither the mechanistic nature of the chaperone defects nor the relevant client substrate are known. In this report, Bhadra et al utilize the yeast prion model as well as multiple functional in vitro assays relevant to DNAJB6 interactions with Hsp70 to determine that known disease mutations lead to altered prion substrate conformation. Additionally, several of the mutations are found to inhibit DNAJB6 dimerization and enhancement of substrate folding/processing by Hsp70 in a dominant manner, effectively poisoning the system. These results are interpreted to provide a foothold into therapeutic intervention for LGMD.

It is of critical importance to begin to connect the wealth of data on chaperone function to genetically linked human proteopathic disease. This study is a solid attempt to do that and provides some novel mechanistic insight while raising additional questions. Importantly, the result that eliminating the NEF Sse partially rescues refolding and prion propagation suggests that Hsp70 ATPase cycling is negatively altered in LGMD. However, the study stops short of answering the critical questions of what the relevant client(s) are for DNAJB6 in LGMD and how the different mutations directly lead to DNAJB6 malfunction, lessening the potential impact. In fact the mutants appear to be defective in every aspect

of Hsp40 function investigated, making it hard to pinpoint a specific problem. Additionally, several aspects of the experimental approach and results are confusing and/or flawed as presented.

1. The title suggests that the work was carried out using the actual DNAJB6 chaperone, which was not the case. It should be modified to reflect that the mutations were modeled in the yeast homolog Sis1.
2. The abstract and final model figure both suggest that the results provide an avenue for therapeutic intervention, but this remains entirely speculative at this point and no such mechanisms are mentioned in the discussion much less explored experimentally. The authors should probably strike the sentence from the abstract.
3. Stylistically, it is highly unusual that every figure panel is separately titled within the figure rather than within the legend. This gives the work a less than professional look.
4. Fig. 1B: it is unclear what exactly is happening to the Rnq1 fibers with the F115I allele. Clearly WT Sis1 is restricting fiber growth but the mutant appears to be completely blocking oligomerization of any kind. This seems inconsistent with the rest of the paper, as mentioned below, since F115I binding to Rnq1 is reduced, if anything. Additional control images are also required of the two Sis1 proteins alone to determine what is being contributed by the chaperones in the images versus the prion.
5. Fig. 1C: It is not clear why the authors chose to plot the refolding rates for luciferase as bar charts for each time point. The rates all appear to be similar but total yield is reduced in the Sis1 mutants. This panel also seems out of place as Fig. 2 continues the Rnq1 experiments.
6. Fig. 2 is unclear to non-aficionados of the ade1 suppression assay. A simplified version of the cartoon in Fig S2 as well as a description of the assay would be welcome in the main body of the paper here.
7. It is unclear what Fig. 2B is trying to show that is any different from 2A. Is this necessary?
8. The conformer-specific effects demonstrated in panel 2C are nebulous modest changes in Rnq1 strain distribution that somehow link to conformation, although this reviewer does not believe specific conformations have been ascribed to the different strain types. At the end of the Fig. 2, it seems we can only say that the overall strain distribution is different in the Sis1 mutants, but not much else.
9. The experiments in Fig. 3 analyze Sis1- and Rnq1-co-stimulated Ssa1 hydrolysis with Rnq1 multimers formed at different temperatures. The results are interpreted as reduced stimulatory activity by the mutants by the warmer-formed Rnq1 variants. However, my read of the data is that the mutants remain largely unchanged in all four panels while the Sis1 WT stimulation progressively drops. Moreover, it is unclear why the authors are using these different Rnq1 temperature variants and what specific information they impart.
10. Fig. 3 also suffers from stylistic and procedural problems. The legend and colors used make it very difficult to discern the multiple mutants and controls (this is a problem in nearly every following figure as well). The Y-axis is also inappropriate. What is "RFU"? ATP hydrolysis should be calculated as mol substrate/mol enzyme/min and labeled as such as is always done for Hsp70 ATPase assays.
11. Fig. 4 is also confusingly presented but appears to show that Sis1 mutants are defective in binding two different substrates and also Ssa1. The approach seems also to be somewhat complicated to determine binding efficiency given that the authors have purified proteins in hand. Why not perform SPR or some other more direct assessment of binding affinity? Lastly, what is happening in Fig. 4D? Why is the X axis broken – the scale is identical to all the others? And how can the wild type Sis1 be an actual curve fit?
12. Fig. 5: While the mutants are clearly defective in homo- and hetero-dimerization, the Y-axis of the plots do not make sense. What is the absorbance a percentage of? Can't be wild type since that is also plotted, and generally relative amounts do not include the raw units. These results also call into question the dominant negative nature of the mutants. How can they be poison subunits while also being defective in dimerization?
13. Fig. 6: As performed, this reviewer believes the experiments are not actually titrations but rather replacements of mutant for WT Sis1 and therefore no different than the results in Fig. 3 A. To truly assay what I think the authors want to assay, the concentration of WT Sis1 should be held constant and increasing amounts of the "poison" subunit added to test for inhibition at stoichiometric or superstoichiometric ratios. Even so, only the endpoint refolding yields are necessary, not every time

pont.

14. Fig. 7: The well-trap assay is interesting and strongly suggests that removing Sse1 compensates for the "broken" Sis1. However the results need to be quantified, replicated and statistically treated and plotted to accurately state that.

15. A bigger issue with Fig. 7 is the authors' apparent misunderstanding of the nature of the two Sse1 mutants. K69Q as reported by the Morano laboratory and K69M from the Bukau and Hartl labs has essentially zero impact on Sse1 function as ATP hydrolysis has little role in the Sse1/Ssa1 NEF cycle. It definitely does not delay binding to Hsp70. The G233D mutation eliminates ATP binding by Sse1 and therefore binding to Ssa1, nearly completely. It therefore delays release of ATP from Ssa1 but only because it never binds Ssa1, and is essentially identical to strains lacking SSE1. Something interesting is going on with the elimination of an NEF and the Sis1 mutants, but as presented this figure is not compelling and incorrectly establishes this potentially intriguing result. Δ Sse1 should also be Δ sse1. Fig. 7B is uninterpretable given the overlap of all the plot markers and error bars.

16. Fig. 8: The cartoons are overly complicated and can probably be reduced to 8B alone, which unfortunately also makes the confusing case that nearly everything is wrong with the Sis1/DNAJB6 mutants making it difficult to parse out precisely the defect.

REVIEWER COMMENTS

We thank the reviewers for their insightful comments which were very helpful in improving the manuscript.

Reviewer #1 (Remarks to the Author):

This manuscript by Bhadra et al describes an interesting investigation - how some pathogenic mutations of human DNAJB6 that cause dominant myopathy might alter its binding ability to substrates and its interactions to co-chaperones by using the budding yeast *S. cerevisiae* as a testing system. *S. cerevisiae* contains several yeast prions, including [RNQ+] which is a substrate of Sis1 – an essential J-protein homologous to DNAJB6. A set of myopathy mutations that occurred in DNAJB6 were generated in Sis1 and the authors found that those mutants have compromised functions in client protein binding, denatured protein refolding, and co-chaperone interactions. These findings provide a mechanistic insight into the pathogenic effect of these mutations on Hsp70/40 ATPase process. [RNQ+] can exist in multiple conformers – prion strains and interestingly, the authors found that different Sis1-myopathy mutants can have a very different effect on de novo formation of different [RNQ+] strains, in other word, prion conformer-specific. Although how much the knowledge learned from here can be transformed into the mammalian skeletal muscles remains to be tested, their finding is interesting and potentially important. It may provide new approach to explore potential therapeutics of myopathy along this line of research. I recommend it be accepted by Nature Communication after the following concerns are addressed:

1) The manuscript would be strengthened if the authors show the quality of the Sis-WT and Sis1-mutants (SDS-PAGE) that were used in this study.

An SDS-PAGE gel image showing the purified chaperone proteins used in this study has now been added as Figure S1 and referred to in the text.

2) Although the conclusion that the impact of Sis1 on Rnq1 fiber formation is mainly at the nucleation step might be correct, the authors should present their data of Fig S1A, B, and C in a clearer format. I suggest use a same scale of Y-axis for all temperatures; and/or calculate the rate of Rnq1 fiber elongation (in the presence of seeded Rnq1 fiber) for each mutant in each temperature and compare them in a table. The current form of graphs is too busy to give a clear picture on what is going on.

All of the scales of Y-axis for Figure S1A, B, and C have now been matched. These are now Figures S2A, B, and C. We have also expanded our analysis and calculated the rate of fibril formation at all temperatures and added these data as Figures S2D, E, and F.

3) Line 219 to 220: the sentence “Once the mutants bind substrate, however, they are able to stimulate-Hsp70 activity (Fig 3D)” does not make sense. Need to clarify what evidence to allow drawing such conclusion.

The line referred to here has now been removed.

4) Supp. Fig. 1D – the quality of the figure is questionable – poor resolution and overexposure. I suggest that the authors either provide a better image to show the differences between the samples or remove this figure.

We removed this figure as suggested.

5) Experiments presented in Fig 6 and supp Fig 5 have some designing problems. The amount of Sis1-WT is different in each examined mixture. One should expect to see Ssa1 ATPase activity being reduced when the concentration of the Sis1-WT is decreased even though the total concentration of Sis1 plus mutant Sis1 remains same as both Sis1 mutants were shown to have a significantly less activity. **To make the point that Sis1-WT dimerization is inhibited by a mutant Sis1, one should keep the concentration of Sis1-WT same across the board while increasing the mutant Sis1 concentration. In this design, you also need to include a set of controls: Sis1-WT alone, mutant Sis1 alone for each concentration that will be added to the mixture. Whether a Sis1 mutant inhibits Sis1 WT dimerization can be seen by comparing these treatments.**

We re-did these ATPase assays as suggested by the reviewer. Sis1-WT concentration was kept constant at 0.03uM and we titrated varying concentrations of mutants (F106L and F115I) in the presence of Rnq1 substrates formed at 18°C (Figure 6A and B) and 25°C (Figure S7 A and B).

Reviewer #2 (Remarks to the Author):

In the current MS, the authors have made several Sis-1 mutations that match with some of the DNAJB6 mutations causing Limb-Girdle Muscular Dystrophy (LGMD) and tested their impact on Sis-1 function using both yeast and in vitro data with purified proteins. To me, the take-home message is that the Sis-1 mutants show (partial) loss of function in their “holdase” function to keep substrates in a non-aggregated form as well as in their ability to interact with Hsp70 (Ssa1) to stimulate its ATPase activity.

My conclusion from the data is further that whereas the loss of holdase function

depends much on the type of the G/F mutant used (different in extent for different substrates), the defect on stimulation the ATPase of Ssa1 seems rather similar for the different mutants. The latter, however, is hard to extract from the data as most experiments focus on co-stimulation of substrates plus Sis-1 (mutant) that are obviously confounded by the fact that Sis-1 mutants differently interact with those substrates. So, to confirm this would require more in-depth analysis of the effects of (different concentrations) of Sis-1 and mutants alone on the ATPase of Ssa1.

Indeed, all of the mutants show reduced ability to stimulate Ssa1 ATPase activity and this effect was also conformer specific (in the presence of denatured Rnq1 there was no difference; Fig S4C). We think this is an important finding itself and may impact the way others evaluate chaperone activity and think about chaperonopathies. Conformer-specific impacts on protein misfolding diseases (prion strains, tau, etc.) may also be an important component of chaperonopathies. In this study, we took advantage of this system and a known substrate to investigate the effects of the G/F domain mutants on chaperone activity. We agree that it would be interesting to perform more in-depth analysis of the effects of the mutants on Hsp70 ATPase stimulation, but we plan to perform such studies with DNAJB6 itself in order to gain more insight into LGMDD1 and we don't need a substrate for those studies.

A main concern is further that whereas Sis-1 shares many functional features with DNAJB6 (for which mutation cause LGMD), it is yet unclear that it is a true homologue. Clearly, Sis-1 also share functional features with human DNAJB1. In fact, features of supporting Hsp70-dependent refolding of heat denatured luciferase (one of the substrates used here) has been demonstrated for DNAJB1 but not DNAJB6. **Moreover, AlphaFold predictions, show that Sis-1 shares structurally features of both. So, even though the current data with equivalent LGMD mutation in Sis-1 are very informative, some cautional remarks must be added.**

We agree with this point. While DnaJB1 is clearly more homologous to Sis1 in terms of domain structure, DnaJB6 has also been shown to be functionally similar in terms of its ability to modify Htt aggregation. We have now added cautional remarks in the Discussion section.

Also, the paper suffers from severe real inaccuracies, jargon and sometimes lack of information that make it hard to digest. All figures are rather sloppy: it is a puzzle to find out with symbols belong to which mutants. **All labelling and coloring should be**

improved, and I suggest using clearer and fixed colors/symbols for the wildtype and mutant Sis-1 variant throughout all figures.

We have now improved the labeling and have used uniform colors for Sis1-WT and the mutants for all the figures in the manuscript as appropriate.

However, the paper does contain very interesting data that, if presented better, more cautiously discussed and maybe supplemented by some additional data at places would very valuable to the chaperone and protein homeostasis and disease community. **The findings that both substrate binding and Hsp70 interactions are each (slightly) impaired in these mutant also helps to elucidate the yet unclear role of the G/F rich region in class A and Class B J-domain proteins and help to understand the functional impact related to the recent findings that this region may actually be involved in an (subtle) autoinhibition of these classes of chaperones.**

We appreciate these comments and have included both additional data and cautionary remarks about DnajB6 and Sis1 similarities and differences as appropriate throughout the manuscript. Interestingly, our data do not suggest that these mutants simply remove or reduce such auto-inhibitory activity as they do not appear to increase the interaction with Hsp70.

Specific comments:

- Figure 1 shows that the G/F mutants of Sis-1 have reduced capacities to delay amyloid formation of Rnq1, the F115L mutant showing the largest LOF. It would be helpful to quantify the data (e.g. time to 50% of ThT RFU) and order the mutants (in my view, the order looks like: F115L < D110Δ < F106L < N108L < wt).

We have now measured the time to 50% of ThT fluorescence and have added that data in a graph as inset to Figure 1A. We did find that all the mutants (except D110Δ) took less time to reach 50% of ThT fluorescence.

We have now ranked all of the relevant experimental results and put that together in a paragraph in the discussion section, highlighting how the mutants rank in terms of functionality throughout the Hsp40/70 cycle.

What does confuse me a bit is the EM data: these show complete absence of fibers in the presence of F115L whilst fibers were formed in the presence of wildtype Sis1 (albeit of different structure). I do not know how this can be matched with the ThT data.

We agree with this comment and indeed found that this was somewhat counterintuitive. We hypothesize that different ThT binding sites are present within these morphologically distinct Rnq1 aggregates and suggest this in the manuscript.

Additionally, we cannot neglect the fact that despite its demonstrated utility as an amyloid stain, continued concerns over potential cross-reactivity of ThT have led to several important studies probing its specificity. For examples; the crystal structure of ThT bound to acetylcholinesterase demonstrated that the dye binds in a site formed primarily of α -helices, in striking contrast to cross- β fibrils (*J Am Chem Soc.* 2008;130:7856–7861). ThT has also been shown to bind to a hydrophobic pocket of human serum albumin with comparable affinity to many drug-like molecules (*Spectrochim Acta A Mol Biomol Spectrosc.* 2009;74:94–99). The capacity of ThT to interact with hydrophobic pockets in non-fibrillar proteins may also rationalize its ability to bind in cavities between protomers of insulin fibrils (*J Struct Biol.* 2007;158:358–369, *J Struct Biol.* 2007;159:483–497). All of these examples clearly highlight the importance of the fibril morphology, protein specificity and the importance of binding pocket for ThT fluorescence. Here, we hypothesize that distinct binding site for ThT are present in these different structures. That hypothesis is supported by our protein transformation data as well. We believe that the impact of chaperone mutants is conformer-dependent and this system uniquely affords ways to determine that that is a possibility worth considering with such studies.

The mutants all also have slight defects in assisting Hsp70-dependent luciferase, but the extent here seems similar for all mutants. Actually, these data should include data points immediately after HS to test if the major effect was not on the extent of luciferase inactivation only, as it seems that the refolding kinetics are not much affected. **Also note, that DNAJB6 was found to be unable to assist refolding, whereas DNAJB1 did (illustrating my general remark that some caution must be considered to directly translate findings with Sis-1 to DNAJB6).** Finally the order of loss of function for this endpoint should be somehow provided (as far as I can see, it now ranks as all mutants show a similar defect: F115L = D110delta = F106L = N108L < wt)

While we did not have the t=0 min data for these experiments, we have added a Supplementary figure (S2G) showing the data point t=5 min (which is right after HS). These data address the reviewer's concern that some of the effect could be due to the extent of luciferase inactivation only. These data indicate that there was no significant difference in the refolding kinetics post 5 min of HS and that the extent of luciferase inactivation was similar for all samples.

- Figure 2: The labelling is totally unclear, especially panel B: what samples are shown

here? Again, try to order the mutants in terms of deficiency (in my view: F115L < F106L < wt). Why not also include the 2 other mutants to link this better to the order of deficiency seen in the ThT assay?

Please also note that (to my knowledge) a role for the G/F region in substrate interactions has only been suggested for Sis-1 and DNAJB1 but for DNAJB6 (as yet).

We have now added a detailed description of what samples are shown in Figure 2B in the Figure Legend (Fig 2). Additionally, the description of the assay is included in both the Figure Legend and in the Supplementary Figure Legend 3.

This assay enabled us to determine the changes in overall strain distribution in the presence of LGMDD1 mutants and hence can be ascribed as a deficiency *per se* and that is why we didn't try to include order of LOF or deficiency here.

As the overall distribution of LGMDD1 mutants (in ThT data) was F115L<N108L<F106L<D110<Sis1-WT (as per 50% to ThT RFU data), we chose only two mutants here because we felt that the gross outcome from the data was sufficient to infer a logical conclusion for this class of mutants.

• Figure 3 + Fig S3 are intriguing as they suggest that that the mutants have reduced capacity to stimulate the ATPase of Ssa1. **Whilst the authors state that this effect is only seen when substrates are added, my impression of the data in Fig S3 are that this is not incorrect. And in fact, it would be highly important to better analyse this.** I do know that combined JDP and substrates have larger effects on Hsp70 ATPases than single components only. **To me the data rather suggest that the different RNQ1 substrates are different in their extent of Hsp70 ATPase stimulation (such data should be provided) and that hence this adds up differently with the effects of the Sis-1 mutants (that all may share the same extent of a minor but functionally relevant defect).** Also, given that the Sis-1 variants have different interactions with the substrates make their combine effects on the Sis-1 ATPase hard to interpret.

Indeed, our data show that Sis1-stimulated Hsp70 ATPase is most robust in the presence of Rnq1 fibers formed at 18°C and that Rnq1 monomer affords little ATPase stimulation, while 25°C and 37°C Rnq1 fibers are similar and in between. The LGMD mutants show the most extensive defect (as compared to WT Sis1) when the “best” substrate (18°C Rnq1 fibers) is used. We appreciate the reviewer suggestions and now display the data as uM ATP/uM Hsp70/min and have updated the text accordingly.

Moreover, the raw data as presented are also not easy to evaluate quantitatively: the

authors should **determine rate parameters** to better allow comparison of the various conditions.

We are currently studying more detailed effect of all the mutants on the stimulation of Ssa1 ATPase activity, using different approaches. One of which is by using different concentrations of substrate. As determining rate parameters requires the same, and it is now part of another extensive study we are unable to show the same in this study.

Further, if I am correct, fibers were formed and then added together with Sis-1. As their data in Figure 1/S1 showed that Sis-1 does not work on seeding aggregation, this may not directly be relevant to what is seen in the ThT.

Yes, indeed the reviewer is correct in thinking that if the Rnq1 fibers are preformed, there is no effect of Sis1-WT or LGMDD1 mutants on the substrate processing as seen with Figure S2A-F.

- Figure 4: This figure now directly analyses the direct binding of the Sis-1 variants to the 2 substrate investigated in Figure 1/2 (in fact, therefore this figure should probably be best presented directly after these functional data). Again, the data should be analysed to provide a ranking of binding efficiency. As far as I can now see this ranking for Rnq1 (D110delta < F106L < N108L < F115L < wt) and Luciferase (D110delta < F106L < F115L < N108L < wt) do not match with the functional data. **How do the authors explain this? What should we learn from this?**

The individual ranking of the assay is now been provided together in the discussion section.

Figure 3 in our manuscript showed the difference in Sis1/mutants stimulated Ssa1 ATPase activity. Subsequently, we wanted to understand the possible mechanistic reason behind these results and thus the figures that followed (including Figure 4) addresses the deficiencies of the mutants.

F115I has the biggest effect on confirmation of substrates (Figures 1 and 2). Given that the presence of Ssa1 reduces mutant-specific effect on binding (Figure 4D and S5), we hesitate to draw conclusions of data that include or do not include Ssa1 in terms of ranking mutant severity.

In this figure binding to ssa1 is also include, which should be more directly compared to the ATPase data in Figure 3 and should also include data on the binding of the Sis-1 without substrate. **What is striking to me here that, at least in the presence of substrates all Sis-1 mutants bind LESS to Hsp70, with almost no difference between them. First, how do we need to interpret this with regards to the data in**

figure 3? Second, would one not expect that, if the G/F mutants would release the auto-inhibitory loop on Hsp70 accessibility that it should be more to Ssa1? This is also why data regarding binding between Ssa1 and Sis-1 mutants without the substrates seem essential here.

The data for binding of Sis1-WT and mutants without substrate is now shown as Figure 4C.

The ATPase assay with monomer showed no difference (Supplementary Figure 4C) and that is the same substrate we use for Figure 4. The ATPase assay with the aggregates (Figure 4A) is similar to the substrate used in ThT assay because that is a seeded fiber formation assay.

The decrease in ATPase activity (in Figure 3) could be, at least in part, due to reduced binding to Hsp70. We have added this in the text.

Our data suggest that the LGMDD1 G/F domain mutants are not simply releasing the auto-inhibitory mechanism, however, as they show reduced binding to Hsp70 in the absence of client. This has been added in the discussion section.

- Figure 5: This data should suggest that the mutant of Sis-1 are impaired in forming a dimer. First, I do not know precisely what exactly is expressed and what it implies: does the mutants still form some dimers? The data should actually include a positive control (CTD deletion mutant lacking the dimerization dimer). Also, what do these data mean in terms of the functionalities described in figure 1-4? Also for that a CTD deletion mutant may be required for comparison in these figure 1-4. **Finally, note that DNAJB6 has not being described as a dimer (rather as polydispersed) and has a rather different CTD than Sisi-1 or DNAJB1 and hence the relevance of this figure for understanding the LMDG-mutants is a bit unclear to me.**

Our data showed that the mutants had reduced self-dimerization efficiency as compared to that with Sis1-WT and Sis1-WT homodimer. Of note, the mutants are functional because they still do dimerize, albeit reduced as compared to the wild-type protein.

We appreciate the suggestion for this control. We created, expressed and purified this protein and added the Sis1- Δ DD control data as Supplementary Figure S6.

The data in Fig.5 indicates a possible explanation for the reduction in chaperone activity observed with these mutants. This is in the manuscript as well (lines 240-242).

We do agree with the reviewer that DNAJB6 has not being described as a dimer (rather as polydispersed) and has a rather different CTD than Sis1 and have added a cautional remark for the same. However, we believe that the decrease dimerization efficiency

observed with the homologous mutants in Sis1 just provides further insight into other possible changes with DNAJB6 G/F domain mutants.

- Figure 6: It is repeatedly written in the MS, in particular in relation to this figure that the mutants are dominant in nature. Whereas it is clear that DNAJB6-related LGMD is indeed dominant, the data on the mutants so far show partial loss of function, which could mean the disease may also be due to haploinsufficiency. The data in figure 6 also do not show, in my view, evidence for dominant negative effects, but only support partial loss of function. To demonstrate dominant-negative effects, one should not use different ratios of wildtype and mutants at a fixed total concentration, but rather use a fixed concentration of the wildtype protein and add increasing concentration of the mutants: if it would result in a bell shaped curve of activity this would suggest dominant negative effects. So, as it stands, I see very little added value of figure 6.

We are thankful to the reviewer for this suggestion to better demonstrate the dominant-negative effects of the mutants. We repeated the entire experiment as suggested by the reviewer (kept the Sis1-WT concentration constant at 0.03 μ M and titrated the same with varying concentrations of mutants (F106L and F115I) in the presence of Rnq1 substrates formed at 18°C (Figure 6A and B) and 25°C (Figure S7 A and B)). We have found that titrating Sis1-WT with LGMDD1 mutants (F106L and F115I) decreased Ssa1 ATPase activity.

- Figure 7: This figure (and its accompanying Fig S7) was presented in a for me very confusing manner. First of all, the data factually show Rnq1 aggregation and not directly prion propagation, which -I know- is linked and prion jargon, but yet confusing. **The experiments should be explained better and also the figure is totally confusing (panel A, upper and lower panel are not even explained). What the data show (as far as I could follow....) is that depletion of Sse1 affected prion handling, but this is true for both wildtype and mutants, implying that increased dwell time on Hsp70 may be beneficial for this substrate.**

We have explained the details of the experiments for Figure 7A, B, S9A, B and C in the protein analysis section (methods). We have also quantified, plotted and added statistics for the same.

We did find that deletion of Sse1 affected prion handling of only the mutants and not the wildtype protein and the same has been shown statistically in the graph for Figure 7A, B and C.

Too me, it is also a total overinterpretation of the data to state that Hsp70 activity inhibition provides a potential mechanism for therapeutic intervention for LMGD (line

272-2740) as it seems from the data that both substrate affinity of the Sis-1 mutants as well as interactions with Hsp70 are impaired. So, the machinery as a whole seems mistuned.

We do agree with the reviewer and have deleted the line referred above.

I could not decipher the data in Figure 7B, let alone the related conclusion.

We have changed the Fig.7B (now as Fig. 7C) to barcharts for clear representation of the data and to avoid overlap of all the plot markers and error bars.

- Discussion: The presented model is very confusing and several statements in the text of the discussion seem to be imprecise: E.g.,
- There are no data showing “client processing” is affected by LGMDD1 G/F domain mutants. (lines 311-314).

This claim was from our previously published work (J Biol Chem. 2014 Jul 25; 289(30):21120-30) and has been referenced in the sentence.

Indeed, as stated later, their substrate “holdase” capacity seems affected (lines 339-340)

We have re-phrased this sentence. It now reads as “Putative “*holdase*” activity is not shown in the Hsp40/70-cycle and may well be altered with the LGMDD1 mutants”. This is included in the text (lines 337-338) in the revised manuscript.

- also, the mutants do not “inhibit” Hsp70 ATPase (lines 315-316); they just stimulate and do so less well than wildtype!

We agree with this comment and have updated the manuscript accordingly.

- there was no difference in the ATPase activity between wildtype and mutants (lines 344-345): Sis-1 has no ATPase activity, it stimulates that of Ssa1, and as stated above this activity IS reduced as is clearly evident from Figure S3B. This requires additional investigations though (as stated above)

The text has been corrected to indicate Sis1/mutants stimulated Ssa1 ATPase activity.

- Indeed, the G/F-rich regions of DNAJB1 have been implicated in an autoinhibitory mechanism that regulates its interactions with Hsp70. Note and cite, that this has now

also been suggested to be the case for DNAJB6/8 (as this is where the LGMD mutations are relevant for).

We have noted and cited the same as Reference 50 and 51.

- Lines 369-373: As stated above: I disagree that the current data are indicative of a dominant negative effect.

We have re-done the experiment as suggested (Figure 6) and the data indicated towards the dominant-negative nature of the mutants.

Reviewer #3 (Remarks to the Author):

While a great deal of molecular detail is available regarding how molecular chaperones recognize and interact with substrate/client proteins, especially in vitro, how altered functions lead to disease is far less well understood. The authors of the manuscript under consideration previously established that mapped patient mutations in the locus DNAJB6, encoding a cytosolic Hsp40 chaperone family member, are linked to a myopathy termed Limb-Girdle Muscular Dystrophy Type D1. However, neither the mechanistic nature of the chaperone defects nor the relevant client substrate are known. In this report, Bhadra et al utilize the yeast prion model as well as multiple functional in vitro assays relevant to DNAJB6 interactions with Hsp70 to determine that known disease mutations lead to altered prion substrate conformation. Additionally, several of the mutations are found to inhibit DNAJB6 dimerization and enhancement of substrate folding/processing by Hsp70 in a dominant manner, effectively poisoning the system. These results are interpreted to provide a foothold into therapeutic intervention for LGMD.

It is of critical importance to begin to connect the wealth of data on chaperone function to genetically linked human proteopathic disease. This study is a solid attempt to do that and provides some novel mechanistic insight while raising additional questions. Importantly, the result that eliminating the NEF Sse partially rescues refolding and prion propagation suggests that Hsp70 ATPase cycling is negatively altered in LGMD. However, the study stops short of answering the critical questions of what the relevant client(s) are for DNAJB6 in LGMD and how the different mutations directly lead to DNAJB6 malfunction, lessening the potential impact. In fact the mutants appear to be defective in every aspect of Hsp40 function investigated, making it hard to pinpoint a specific problem. Additionally, several aspects of the experimental approach and results are confusing and/or flawed as presented.

When we initiated this project, we had no ideas about the relevant client(s) for DNAJB6 in LGMDD1. We have explored multiple z-disc proteins as potential clients recently. However, due to genetically heterogeneous nature of these myopathies, it was very difficult to select a particular client substrate for LGMDD1. In last few years, along with the study we have submitted here, we have worked in parallel towards identifying the important client substrates for DNAJB6 and have now potentially identified a few. However, further analyses are still pending to confirm those candidates. Thus, a more extensive analysis with DNAJB6 and its client substrates is beyond the scope of this study. We are indeed hopeful that such will be presented in our future work.

Additionally, we have addressed most of the queries raised by the reviewers and have made significant improvements to the manuscript accordingly.

1. The title suggests that the work was carried out using the actual DNAJB6 chaperone, which was not the case. It should be modified to reflect that the mutations were modeled in the yeast homolog Sis1.

In the revised manuscript we have added cautional remark regarding directly translating the findings with Sis1 to DNAJB6 given their differences (see change in the title and response to reviewer #2).

2. The abstract and final model figure both suggest that the results provide an avenue for therapeutic intervention, but this remains entirely speculative at this point and no such mechanisms are mentioned in the discussion much less explored experimentally. The authors should probably strike the sentence from the abstract.

We have now rephrased the sentence and have added that these results provide initial clues for future therapeutic interventions.

3. Stylistically, it is highly unusual that every figure panel is separately titled within the figure rather than within the legend. This gives the work a less than professional look.

We have removed all the figure headings.

4. Fig. 1B: it is unclear what exactly is happening to the Rnq1 fibers with the F115I allele. Clearly WT Sis1 is restricting fiber growth but the mutant appears to be completely blocking oligomerization of any kind. This seems inconsistent with the rest of the paper, as mentioned below, since F115I binding to Rnq1 is reduced, if anything. Additional control images are also required of the two Sis1 proteins alone to determine what is being contributed by the chaperones in the images versus the prion.

The authors agree with the reviewer that the results observed with F115I mutant in Figure 1B are indeed counterintuitive. We believe that different ThT binding site is present within these morphologically distinct Rnq1 aggregates. This has now been addressed in the manuscript.

We have now added the relevant control TEM images as requested.

5. Fig. 1C: It is not clear why the authors chose to plot the refolding rates for luciferase as bar charts for each time point. The rates all appear to be similar but total yield is reduced in the Sis1 mutants. This panel also seems out of place as Fig. 2 continues the Rnq1 experiments.

We agree with the reviewer that the rate of refolding kinetics was similar and that the total yield was reduced in the Sis1 mutants. That was the reason we didn't claim that the refolding kinetics was different. At the same time, we only mentioned that the mutants were compromised in refolding heat-denatured luciferase as indicated by their low yield.

We chose to show the graph as bar charts because of the overlapping lines (with error bars) making it hard to visualize. That was also the case with Figure 7C (which is now been changed to bar graph as well).

In Figure 1 (sections), we sought to determine the effect of these mutants on substrate processing in general and thus used two different substrates (one prion substrate-Rnq1 and the other non-prion substrate- luciferase) and two different assays to address the same. As we moved to further investigation with respect to the underlying mechanism, we focused on using different conformers of one client. We agree with the reviewer that Figure 2 continues with the Rnq1 experiments, we believe that this was the most appropriate section to show the effect with a different client.

6. Fig. 2 is unclear to non-aficionados of the ade1 suppression assay. A simplified version of the cartoon in Fig S2 as well as a description of the assay would be welcome in the main body of the paper here.

A simplified cartoon is now added as Figure 2A. The detailed assay is diagrammatically represented as Fig S3A. The detailed description of the assay is in the figure legend (Figure 2) and also in Supplementary Figure S3A. The more detailed explanation has been added in the figure legend due to space constraints.

7. It is unclear what Fig. 2B is trying to show that is any different from 2A. Is this necessary?

Figure 2B shows the different Rnq1 strains based on growth, color and prion maintenance (in SD-Ade/YPD/GdnHCl plates). This figure basically represents the phenotypic basis (for what colonies were counted) for the quantification data shown in Figure 2C.

Figure 2A has now been placed as Supplementary Figure S3B. These data represent the yeast strains expressing the RRP reporter displaying different levels of nonsense suppression in [RNQ+] variants. This matches our previously characterized [RNQ+] variants and has been used to determine changes with the addition of LGMDD1 mutants. We included these data as controls and moved them to the Supplement.

8. The conformer-specific effects demonstrated in panel 2C are nebulous modest changes in Rnq1 strain distribution that somehow link to conformation, although this reviewer does not believe specific conformations have been ascribed to the different strain types. At the end of the Fig. 2, it seems we can only say that the overall strain distribution is different in the Sis1 mutants, but not much else.

We agree with the reviewer and have now changed the conclusion from this figure. We have included in the text that the difference in the distribution of [RNQ+] strains generated in the presence of LGMDD1 mutants suggest that amyloid structures can be altered by mutant chaperones.

9. The experiments in Fig. 3 analyze Sis1- and Rnq1-co-stimulated Ssa1 hydrolysis with Rnq1 multimers formed at different temperatures. The results are interpreted as reduced stimulatory activity by the mutants by the warmer-formed Rnq1 variants. However, my read of the data is that the mutants remain largely unchanged in all four panels while the Sis1 WT stimulation progressively drops. Moreover, it is unclear why the authors are using these different Rnq1 temperature variants and what specific information they impart.

In our previous work, we found client-conformer specific defects *in vivo* with the LGMDD1 mutants (J Biol Chem. 2014 Jul 25; 289(30):21120-30). Therefore, in order to conduct these analyses with our previous results in mind we wanted to determine whether the client conformation impacted our ability to elucidate the function of the mutants.

As we know from our previous work that incubating Rnq1 at different temperatures generates different [RNQ+] variants, the idea of using them in the ATPase assays was to show how the different Rnq1 substrate conformers differed in extent of client processing and Hsp70 ATPase stimulation. Consequently, we see that the Sis-1 mutants show a similarly minor, but functionally relevant, defect.

10. Fig. 3 also suffers from stylistic and procedural problems. The legend and colors used make it very difficult to discern the multiple mutants and controls (this is a problem in nearly every following figure as well). The Y-axis is also inappropriate. What is “RFU”? ATP hydrolysis should be calculated as mol substrate/mol enzyme/min and labeled as such as is always done for Hsp70 ATPase assays.

We have now changed all of the colors representing the Sis1-WT and LGMDD1 mutants and have made them same across all figures. The graphs in Figure 3A-C are now presented in the bar chart format for enhanced clarity. ATP hydrolysis has been calculated and is presented as $\mu\text{M ATP}/\mu\text{M Hsp70}/\text{min}$ for all.

11. Fig. 4 is also confusingly presented but appears to show that Sis1 mutants are defective in binding two different substrates and also Ssa1. The approach seems also to be somewhat complicated to determine binding efficiency given that the authors have purified proteins in hand. Why not perform SPR or some other more direct assessment of binding affinity? Lastly, what is happening in Fig. 4D? Why is the X axis broken – the scale is identical to all the others? And how can the wild type Sis1 be an actual curve fit?

We followed the methods from previous important work by Aron et.al. (*Genetics*. 2005 Apr; 169(4):1873-82) for these experiments. We had expertise with ELISA and believe that it is still a standard technique to analyze binding deficiencies. We did not have expertise with SPR or other techniques for this and are confident that the conclusion is valid and can be compared to previous studies with this method.

We have re-graphed Figure 4D (shown as Supplementary Figure 5 in the revised manuscript) and Sis1 does show a perfect fit. The X-axis is broken at the end to show the fit.

12. Fig. 5: While the mutants are clearly defective in homo- and hetero-dimerization, the Y-axis of the plots do not make sense. What is the absorbance a percentage of? Can't be wild type since that is also plotted, and generally relative amounts do not include the raw units. These results also call into question the dominant negative nature of the mutants. How can they be poison subunits while also being defective in dimerization?

We agree with the reviewer and have now represented data as Absorbance at 450nm. This representation is similar to that in Aron et.al. (*Genetics*. 2005 Apr; 169(4):1873-82).

Our data show that the mutants have reduced dimerization efficiency to self and also to Sis1-WT as compared to Sis1-WT homodimer. Of note, that the mutants are functional because they still do dimerize, albeit less as compared to the wild-type protein.

13. Fig. 6: As performed, this reviewer believes the experiments are not actually titrations but rather replacements of mutant for WT Sis1 and therefore no different than the results in Fig. 3 A. To truly assay what I think the authors want to assay, the concentration of WT Sis1 should be held constant and increasing amounts of the “poison” subunit added to test for inhibition at stoichiometric or superstoichiometric ratios. **Even so, only the endpoint refolding yields are necessary, not every time point.**

We appreciate this suggestion and re-did the entire ATPase assay experiment as suggested by the reviewer. We kept the Sis1-WT concentration constant at 0.03uM and titrated the same with varying concentrations of mutants (F106L and F115I) in the presence of Rnq1 substrates formed at 18°C (Figure 6A and B) and 25°C (Figure S7 A and B).

14. Fig. 7: The well-trap assay is interesting and strongly suggests that removing Sse1 compensates for the “broken” Sis1. However the results need to be quantified, replicated and statistically treated and plotted to accurately state that.

We have now quantified, plotted and added statistics as suggested by the reviewer.

15. A bigger issue with Fig. 7 is the authors’ apparent misunderstanding of the nature of the two Sse1 mutants. K69Q as reported by the Morano laboratory and K69M from the Bukau and Hartl labs has essentially zero impact on Sse1 function as ATP hydrolysis has little role in the Sse1/Ssa1 NEF cycle. It definitely does not delay binding to Hsp70. The G233D mutation eliminates ATP binding by Sse1 and therefore binding to Ssa1, nearly completely. It therefore delays release of ATP from Ssa1 but only because it never binds Ssa1, and is essentially identical to strains lacking SSE1. **Something interesting is going on with the elimination of an NEF and the Sis1 mutants, but as presented this figure is not compelling and incorrectly establishes this potentially intriguing result.** Δ Sse1 should also be Δ sse1. Fig. 7B is uninterpretable given the overlap of all the plot markers and error bars.

We are grateful to the reviewer for these comments. After extensive literature search and revisiting our data with the Sse1-mutants; we decided to delete the data with K69Q mutant. We have kept the data with G233D and have now stated that it served as an

additional control to the deletion of *Sse1* (as this mutation is indeed similar to strains lacking *Sse1*, as pointed by the reviewer).

We agree with the reviewer that the elimination of NEF has a positive impact on the *Sis1* mutants and have highlighted this in the text. We are doing further experiments with NEF and these mutants and hope to delve into more details in future work.

We have changed $\Delta Sse1$ to $\Delta sse1$ throughout the manuscript.

We have changed the Fig.7B (now as Fig. 7C) to be represented in bar charts for clearer representation of the data (and to avoid overlap with the error bars).

16. Fig. 8: The cartoons are overly complicated and can probably be reduced to 8B alone, which unfortunately also makes the confusing case that nearly everything is wrong with the *Sis1/DNAJB6* mutants making it difficult to parse out precisely the defect.

We agree with the reviewer and have kept only Figure 8B as the main Figure 8. We have shifted Figure 8A to Supplementary Figure 10 as it simply shows the normal functioning of Hsp70/40 cycle.

We agree with the reviewer that from the first half of the *Ssa1* ATPase cycle; *Sis1/DNAJB6* mutants showed reduced dimerization, binding with substrate, and Hsp70 and that nearly everything was wrong. Thus the inhibition of the ATPase cycle is quite possibly due to the loss of function of these mutants. Our results highlight that every aspect of the functionality of Hsp40/70-cycle is impacted by the disease-causing missense mutations in the G/F domain. We have highlighted this along with the analysis of the severity of the mutants across various assays in the discussion section (lines 306-316).

Reviewers' Comments:

Reviewer #1:

Remarks to the Author:

The authors have fully addressed my concerns. I am satisfied with the revision and think the revised article is ready for publication in Nature Communications.

Reviewer #2:

Remarks to the Author:

Overall, the authors replied well to some of my comments, I do still have residual concerns.

- In their reply to my question on presumed defects of the mutants regarding stimulation of the ATPase of Ssa1, they state this is an important finding itself and may impact the way others evaluate chaperone activity and think about chaperonopathies. I do not disagree, but my question was that I wondered whether this is due to an impaired interaction of the mutant Sis-1 with HSP70 (direct) or merely a result of the fact that the mutants have an impaired substrate-interaction, hence affecting their co-stimulatory effect on the ATPase of Hsp70. I think this is relevant in the context of the data on the role of a CTD-G/F interaction within these classes of JDP proteins impeding on a second JDP-Hsp70 interaction. It seems not too complex to directly measure this with the Sis-1 mutants they have (rather than saying they plan to do so with DNAJB6).

I truly think this is important also; the current assumptions are that the negative auto-inhibitory (CTD-G/F) is directly affecting Hsp70 interactions. Their finding that mutations in the G/F region directly affect substrate binding is indeed truly novel. If they would find that without substrate the G/F mutants would normally interact with and stimulate Hsp70, this would be very interesting and could imply hereby interaction (of the complex) with Hsp70 is driven by the Sis-1-substrate interaction

- Also, some of the (largely cosmetic) changes now mention that all these data concern Sis-1 and not DNAJB6, this is (to me) insufficient. Even more so, in the abstract, they still write "dominant mutations in DNAJB6 (Hsp40/Sis-1)" as well as that they "determined how LGMDD1 mutations affect Hsp40 function". Their title even states Hsp40, which is incorrect (DNAJB6 is not even 40kDa and for mammals Hsp40 refers to DNAJB1). This is (a still cool) paper on Sis-1, but things should not be overstated (for translational relevance). It also does not suffice to only address this with a single (not very) cautionary remark at the end of the discussion.

- The response to my comment on the disconnection between the ThT data and EM data is well appreciated but my questions still remain. For ThT (figure 1, panel A), the Sis-1 mutants show much less delay ThT binding than the wildtype sis-1 (a bit strange that the inset does not reflect this...), whereas for the EM (figure 1, panel B), the F115I mutant (unclear why only this one was tested here) has a much bigger effect than the wildtype. So, panel B shows LOF of the mutants (conclusion in lines 131-132), but panel B suggests improved function (with a newly added conclusion in lines 141-143 on different ThT binding sites that I do not fully understand).

- Regarding my concerns about the effects on luciferase refolding, I am also not fully satisfied. First: why provide these in S2G and why not add these 5 min time points into Figure 1C (separately there are hard to compare). In addition, I tried to calculate halftimes of recovery from the data in figure 1 (which is, of course, inaccurate), but this was maximally 20% lower (significant?) for all mutants with the major difference being the extrapolation of activity to 0 hours after treatment (about 2-fold lower than for the wildtype). I think, proper kinetics are needed here to distinguish between a 'holdase' effect (maybe revealing a defect in substrate interaction) or refolding effect (related to altered efficacy to interact with Hsp70) of the mutants. Not sure also why such an experiment, including a time point immediately after heat shock, cannot be included.

- The responses related to DNAJ dimerization are also a bit cosmetic, rather than to the point. A statement that most DNAJ proteins oligomerize in some way is not only vague but also incorrect. We only know that some can (in vitro) but whether they truly do so in vivo is unclear. True dimerization has only been demonstrated for a few (indeed including Sis-1, but never shown for DNAJB6 which actually lacks this DD-domain! So, the relevance of any of these data to the LGMD mutants of DNAJB6 is yet unclear. I further appreciate the authors included the CTD deletion mutant for showing their dimerization analyses, but to make the point of the relevance of dimerization for the G/F mutants, it should also of course have been included in the functional data.
- I much appreciate the added experiments in Figure S7) indeed demonstrating a dominant-negative effect of the mutants with regards to stimulating the HSP70 ATPase.
- The model in figure 8 is still vague to me and suggestion for therapeutic route should not be given (also not in abstract) since all the work still concerns Sis-1 only.

Reviewer #3:

Remarks to the Author:

The manuscript has been substantially revised to address all of my major concerns, and those of the other two reviewers by my assessment. Less well-supported claims are softened, and experimental issues have been properly addressed.

The primary impact remains the thorough characterization of several disease-linked mutations in Sis1 modeled on those found in DNAJB6. This work breaks new ground in understanding this particular chaperonopathy and should be of interest to a broad readership.

Reviewer #2 (Remarks to the Author):

Overall, the authors replied well to some of my comments, I do still have residual concerns.

- In their reply to my question on presumed defects of the mutants regarding stimulation the ATPase of Ssa1, they state this is important finding itself and may impact the way others evaluate chaperone activity and think about chaperonopathies. I do not disagree, but my question was that I wondered whether this is due to an impaired interaction of the mutant Sis-1 with HSP70 (direct) or merely a result of the fact that the mutants having an impaired substrate-interaction, hence affecting their co-stimulatory effect on the ATPase of Hsp70. I think this is relevant in the context of the data of the role of a CTD-G/F interaction within these classes on JDP proteins impeding on a second JDP-Hsp70 interaction. It seems not too complex to directly measure this with the Sis-1 mutants they have (rather than saying they plan to do so with DNAJB6).

I truly think this is important also; the current assumptions are that the negative auto-inhibitory (CTD-G/F) is directly affecting Hsp70 interactions. Their finding that mutations in the G/F region directly affect substrate binding is indeed truly novel. If they would find that without substrate the G/F mutants would normally interact with and stimulate Hsp70, this would be very interesting and could imply hereby interaction (of the complex) with Hsp70 is driven by the Sis-1-substrate interaction

We appreciate the reviewer's comment, and also point out that Supplementary Figure 4B data show that there was no difference in the Sis1-WT/mutants stimulated ATP-hydrolysis rate of Hsp70 in the absence of client protein. We have now performed and added experimental data in the absence of the client protein using varying concentrations of Sis1-WT/mutants (as suggested by this reviewer). We have again observed that Sis-WT/mutants stimulated Hsp70 ATPase activity is unaltered in the absence of the client protein. This indicates that the complex (Sis1-substrate-Hsp70) formed with Hsp70 is indeed driven by the Sis1-substrate interaction. This is now as the new Supplementary Figure 4B.

- Also, some of the (largely cosmetic) changes now mention that all these data concern Sis-1 and not DNAJB6, this is (to me) insufficient. Even more so, in the abstract, they still write "dominant mutations in DNAJB6 (Hsp40/Sis-1)" as well as that they "determined how LGMDD1 mutations affect Hsp40 function". Their title even states Hsp40, which is incorrect (DNAJB6 is not even 40kDa and for mammals Hsp40 refers to DNAJB1). This is (a still cool) paper on Sis-1, but things should not be over-stated (for translational relevance). It also does not suffice to only address this with a single (not very) cautional remark at the end of the discussion.

There have conflicting reports about Sis1 being the homologue of DNAJB6 and we fully appreciate this. However, recent studies attempting to understand DNAJB6 function using other substrates like Huntingtin protein (polyQ) also use Sis1 in a yeast system (Nat Commun. 2020 Dec 8;11(1):6271. doi: 10.1038/s41467-020-20000-x), with caution of course. We do not agree with the reviewer that DNAJB6 is not an Hsp40 protein (some refer to these as J-proteins knowing that these have varying molecular weights, however, we cannot ignore the fact structurally DNAJB6 classically resembles canonical Type II Hsp40 proteins) and we know that

in mammals Hsp40 refers to only DNAJB1. We think what the reviewer wanted to state here is that Sis1 is more homologous to human DNAJB1 and not B6 and we agree with that, as do others. At the same time, one cannot ignore the fact that even DNAJB1 which is presumed to be the most appropriate homologue of Sis1, differs in terms of the G/F domain region, which makes it difficult to estimate the degree of the J-domain release in the Sis1 variants based on DNAJB1 (PNAS 2021 Vol. 118 No. 49 e2108163118).

We were really successful in translating our findings using yeast Sis1 (Hsp40) to understand muscle chaperonopathy concerning DNAJB6 protein through our previous studies [(J Clin Invest. 2020 Aug 3;130 (8):4470-4485. doi: 10.1172/JCI136167) and (J Biol Chem. 2014 Jul 25;289(30):21120-30. doi: 10.1074/jbc.M114.572461)]. Thus, we don't believe that we have over-stated anything in terms of translational relevance. Moreover, we have toned down our therapeutic claims at this point with these data, however, we cannot ignore the fact that these data thus provide a crucial direction to further look into for therapeutic interventions for these myopathies. In order to address these concerns, and at the same time not undermine the relevance of this study, we decided to change the title of this study by incorporating Hsp40 instead of DNAJB6 (this was also in agreement with Reviewer 3).

Additionally, we have provided cautional remarks at other places (e.g.; where we talked about dimerization; lines 359-362) instead of only providing it only at the end of the discussion section (lines 402-406).

- The response to my comment on the disconnection between the ThT data and EM data is well appreciated but my questions still remains. For ThT (Figure 1, panel A), the Sis-1 mutants show much less delay ThT binding that the wildtype sis-1 (a bit strange that the inset do not reflect this...), whereas for the EM (figure 1, panel B), the F115I mutant (unclear why only this one was tested here) has a much bigger effect that the wildtype. So, panel B shows LOF of the mutants (conclusion in lines 131-132), but panel B suggest improved function(with a newly added conclusion in lines 141-143 on different ThT binding sites that I do not fully understand).

We understand the concern the reviewer has shown regarding the ThT and the EM-data. However, the most notable point here is the difference in the experimental samples which have been used in these experiments. For the ThT assay (shown in Figure 1A), Rnq1 monomer was used as a client protein (with no preformed Rnq1 fibers) and in this kinetic assay, monomeric Rnq1 was allowed to form aggregates/fibers with beads (which have shown to promote aggregate formation) and constant shaking of the microtiter plates in the presence of +/- chaperones with ThT was done at Room Temperature (23°C) in the spectrophotometer. On the other hand, for the EM-data, we generated Rnq1 fibers of particular conformer (by incubating the same at 18°C) in the presence or absence of chaperones. As such, it is important to point out that technically the client for these experiments is not exactly similar in terms of their conformations. The EM-data in Figure 1B is more comparable to data in Figure 2C in terms of similarity of the substrate used. And as evident from our data in Figure 2C, where we clearly see that in the presence of Sis1-F115I, there was a significant difference in the distribution of [RNQ+] variants as compared to Sis1-WT. Therefore, the presence of distinct conformers of Rnq1 substrate with unique ThT binding sites cannot be ruled out and is in fact, likely.

- Regarding, my concerns about the effects on luciferase refolding, I am also not fully satisfied. First: why provide these in S2G and why not add these 5 min time points into Figure 1C

(separately there are hard to compare). In addition, I tried to calculate halftimes of recovery from the data in figure 1 (which is, of course inaccurate), but this were maximally 20% lower (significant?) for all mutants with the major difference being the extrapolation of activity to 0 hours after treatment (about 2-fold lower than for the wildtype). I think, proper kinetics are needed here to distinguish between a 'holdase' effect (maybe revealing a defect in substrate interaction) or refolding effect (related to altered efficacy to interact with Hsp70) of the mutants. Not sure also why such an experiment, including a time point immediately after heat shock, cannot be included.

We have provided 5 min data as separate graph because it was not the part of the same experiment. It was done separately (and in replicates) where we found that the assay time points needed to go beyond 15 minutes to see a difference in the refolding of luciferase in the presence of Sis1-WT and the mutants. This is why we don't have any data at 0 min, but the 5 minute time point addresses the point originally raised.

From the data presented in Figure 1C, we wanted to determine how these mutants affect non-prion substrate processing. We do see that the mutants were all compromised in their ability to refold luciferase.

We are not trying to establish any holdase or refolding effect from these data, nor do we mention such in the manuscript. However, we revisited our entire data set and found that even with luciferase as substrate, mutants were defective in interacting with it (Figure 4B) establishing the "holdase" effect. At the same time, there is strong possibility of the existence of a refolding effect also, as evident from our binding data in Supplementary Figure 5 (where we have used varying concentrations of Ssa1 (0-100nm) and found that all mutants were defective in binding to Hsp70 in luciferase-bound form.

More data would be needed to calculate halftimes of refolding, but we do not believe that having this information adds or changes the conclusions.

- The responses related to DNAJ dimerization are also a bit cosmetic, rather than to the point. A statement that most DNAJ proteins oligomerize in some way is not only vague but also incorrect. We only know that some can (in vitro) but whether they truly do so in vivo is unclear. True dimerization has only been demonstrated for a few (indeed including Sis-1, but never shown for DNAJB6 which actually lacks this DD-domain! So, the relevance of any of these data to the LGMD mutants of DNAJB6 is yet unclear. I further appreciate the authors included the CTD deletion mutant for showing their dimerization analyses, but to make the point of the relevance dimerization for the G/F mutants, it should also of course have been included in the functional data.

We appreciate the reviewer concern regarding using the sentence "Most DNAJ proteins oligomerizes in some way" and also appreciate the fact that true dimerization have been demonstrated only for few Hsp40 proteins (and that there exist debates that some dimerizes in-vitro but not known whether they do so in-vivo). However, most of the reported literature suggests that class I and class II Hsp40 protein exists as dimers (may be oligomers of some sort). And in our case, as mentioned by the reviewer Sis1 do exists as a dimer and DNAJB6 as a polydispersed oligomer. Thus, we have changed the sentence to "Most of the canonical class I and class II J-proteins exist as oligomers".

Secondly, as DNAJB6 exists as polydispersed oligomers, it is really difficult to understand its interactions with peptides/clients. People have combined homology modelling, chemical crosslinking mass spectrometry, small-angle X-ray scattering (SAXS) and negative stain electron microscopy (EM) to obtain a structural model of the monomer, dimer and higher order oligomeric forms of DNAJB6 [(Scientific Reports volume 8, Article number: 5199 (2018)]. The dimer model reveals a peptide-binding cleft lined with S/T-residues located at the interface between the two monomers in the DNAJB6 dimer. Detection of ¹⁴N-¹⁵N hybrid crosslinks shows that the DNAJB6 oligomers are dynamic entities that can exchange subunits and SAXS data indicate well folded DNAJB6 oligomers, forming a heterogeneous population with varying number of subunits, into which the structural model of the DNAJB6 dimers fit. Another study has revealed that the CTD (but not the J domain) of DNAJB6 self-associates to form an oligomer comprising ~35 monomeric units, revealing an intricate balance between intramolecular and intermolecular interactions (Proc Natl Acad Sci U S A. 2019 Oct 22;116(43):21529-21538. doi: 10.1073/pnas.1914999116). The point of discussing this here is to establish the fact that oligomers are made of multiple monomeric units (some maybe dimers or combination of two or more dimers) of the protein and presently, it is still a debatable issue that requires further investigation. Thus, although not directly consequential, however, the importance of Sis1 dimerization in terms of these mutants and much into DNAJB6 cannot be ignored.

We have deleted the dimerization domain of Sis1 and have used that as a control in the dimerization assay. We have now used the same Sis1- Δ DD mutant in the ATPase assay (without client) and found that there is no difference in the Sis1- Δ DD mediated Hsp70 stimulation of ATPase activity (Supplementary Figure 4B) as compared to both Sis1-Wt/mutants. This was not surprising as the question of release of G/F domain autoinhibition doesn't arise when using this control (Sis1- Δ DD) as that is regulated through the interaction of Sis1 CTDI with EEVD motif of Hsp70. Thus, we didn't see the relevance of using this in other functional assays.

- I much appreciate the added experiments in Figure S7) indeed demonstrating a dominant-negative effects of the mutants with regards to stimulating the HSP70 ATPase.\

Thanks for the comment.

- The model in figure 8 is still vague to me and suggestion for therapeutic route should not be given (also not in abstract) since all the work still concerns Sis-1 only.

We are in agreement with the reviewer in terms of his concern that this study is still been conducted using only Sis1. However, we are optimistic that these findings are translatable in some way. As mentioned above, we were able to translate our findings with Sis1 to show similar results with mammalian DNAJB6 protein in our previous studies.

Moreover, as mentioned in our manuscript we are not claiming that these data point to the exact therapeutic route to follow, but a crucial direction to begin to explore for therapeutic interventions for this myopathy (especially when there are still debates about the exact clients for DNAJB6 protein in terms of muscle pathology).